

**SM2RAIN-ASCAT (2007-2018): global daily satellite rainfall**
**from ASCAT soil moisture**
Luca Brocca[1*], Paolo Filippucci[1], Sebastian Hahn[2], Luca Ciabatta[1], Christian Massari[1],
Stefania Camici[1], Lothar Schüller[3], Bojan Bojkov[3], Wolfgang Wagner[2]
[1]{Research Institute for Geo-Hydrological Protection, National Research Council, Perugia, Italy}
[2]{Department of Geodesy and Geoinformation, Vienna University of Technology, Vienna, Austria }
[3]{European Organisation for the Exploitation of Meteorological Satellites, Darmstadt, Germany}

* Correspondence to: Ph.D. Luca Brocca, Research Institute for Geo-Hydrological Protection, National Research Council, Via della Madonna Alta 126, 06128 Perugia, Italy. Tel: +39 0755014418 Fax: +39 0755014420 E-mail: luca.brocca@irpi.cnr.it.



## Abstract

Long-term gridded precipitation products are crucial for several applications in
hydrology, agriculture and climate sciences. Currently available precipitation products obtained
from rain gauges, remote sensing and meteorological modelling suffer from space and time
inconsistency due to non-uniform density of ground networks and the difficulties in merging
multiple satellite sensors. The recent "bottom up" approach that uses satellite soil moisture
observations for estimating rainfall through the SM2RAIN algorithm is suited to build long-
term and consistent rainfall data record as a single polar orbiting satellite sensor is used.
We exploit here the Advanced SCATterometer (ASCAT) on board three Metop satellites,
launched in 2006, 2012 and 2018. The continuity of the scatterometer sensor on European
operational weather satellites is ensured until mid-2040s through the Metop Second Generation
Programme. By applying SM2RAIN algorithm to ASCAT soil moisture observations a long-
term rainfall data record can be obtained, also operationally available in near real time. The
paper describes the recent improvements in data pre-processing, SM2RAIN algorithm
formulation, and data post-processing for obtaining the SM2RAIN-ASCAT global daily rainfall
dataset at 12.5 km sampling (2007-2018). The quality of SM2RAIN-ASCAT dataset is assessed
on a regional scale through the comparison with high-quality ground networks in Europe,
United States, India and Australia. Moreover, an assessment on a global scale is provided by
using the Triple Collocation technique allowing us also the comparison with other global
products such as the latest European Centre for Medium-Range Weather Forecasts reanalysis
(ERA5), the Global Precipitation Measurement (GPM) mission, and the gauge-based Global
Precipitation Climatology Centre (GPCC) product.
Results show that the SM2RAIN-ASCAT rainfall dataset performs relatively well both
at regional and global scale, mainly in terms of root mean square error when compared to other
datasets. Specifically, SM2RAIN-ASCAT dataset provides better performance better than
GPM and GPCC in the data scarce regions of the world, such as Africa and South America. In
these areas we expect the larger benefits in using SM2RAIN-ASCAT for hydrological and
agricultural applications.
The      SM2RAIN-ASCAT      dataset      is      freely      available      at
https://doi.org/10.5281/zenodo.2591215.
*Keywords:* Rainfall, Soil moisture, ASCAT, SM2RAIN, Remote Sensing.



# 1 Introduction

Rainfall is ranked the first among the Essential Climate Variable by the Global Climate Observing System (GCOS) as it represents the most important variable in many applications in geosciences (Maggioni and Massari, 2018). Near real-time rainfall data are needed for the mitigation of the impacts of natural disasters such as floods and landslides (e.g., Wang et al., 2107; Camici et al., 2018; Brunetti et al., 2018; Kirschbaum and Stanley, 2018) while long-term rainfall record are essential for drought monitoring (e.g., Forootan et al., 2019), water resources management (e.g., Abera et al., 2017) and climate studies (e.g., Herold et al., 2016; Pendergrass and Knutti, 2018). Additional applications in which rainfall plays a crucial role are weather forecasting, agricultural planning, vector-borne and waterborne diseases (e.g., Rinaldo et al., 2012; Thaler et al., 2018).

Three different techniques can be used for estimating rainfall: ground measurements, meteorological modelling and remote sensing. Ground measurements are based on rain gauges and meteorological radars (Lanza et al., 2009), but also new approaches such as microwave links are being developed (e.g., Overeem et al., 2011). These measurements guarantee high accuracy but suffer in many regions from limited spatial coverage (Kidd et al., 2017). Alternatively, meteorological models (e.g., reanalysis) are used to estimate rainfall mainly in areas without ground reliable observations (Ebert et al., 2007). The uncertainties associated with these estimates can be large, mainly in areas where ground observations are scarce (Massari et al., 2017a). Therefore, to fill the gaps in the spatial coverage of ground measurements, and to improve the estimates obtained by models, different remote sensing techniques have been developed in the last 30 years (Hou et al., 2014). The standard methods for measuring rainfall from space are based on instantaneous measurements obtained from microwave radiometers, radars, and infrared sensors (Kidd and Levizzani, 2011). These methods are based on the inversion of the atmospheric signals reflected or radiated by atmospheric hydrometeors, i.e., a "top down" approach (Brocca et al., 2014).

The most recent and successful example of satellite precipitation estimates is represented by the Integrated Multi-Satellite Retrievals for GPM (IMERG) of the Global Precipitation Measurement (GPM) mission (Hou et al., 2014) which provide high spatial (0.1°) and temporal (30-minute) resolution and quasi-global coverage (+/-60°). To obtain such resolution and coverage, the IMERG products use a constellation of polar and geostationary satellite sensors operating in the microwave and infrared bands. However, the use of multiple sensors has some



problems, including: the inconsistency between rainfall estimates from different sensors
(intercalibration problem), the difficulties in collecting observations from multiple space
agencies (i.e., problem of delivering the products in near real-time), and the high costs for the
operation and the maintenance of the overall constellation. Moreover, as the top down approach
requires the merging of instantaneous rainfall measurements from multiple sensors, the failure
of one of them may imply a significant degradation in the accuracy of accumulated rainfall
estimate due to the high temporal variability of rainfall (Trenberth and Asrar, 2014).
In recent years, a new "bottom up" approach has emerged that uses satellite soil moisture
observations to infer, or to correct, rainfall over land (Brocca et al., 2013a; Crow et al., 2009;
Pellarin et al., 2013; Wanders et al., 2015). The major difference between the bottom up and
top down approaches is in the type of measurement; i.e., accumulated rainfall with the bottom
up method and instantaneous rainfall rates with the top down method. This difference makes
the two approaches highly complementary and their integration has been already successfully
tested and demonstrated in several recent studies (e.g., Brocca et al., 2016; Ciabatta et al., 2017;
Chiaravallotti et al., 2018; Massari et al. 2019). When accumulated rainfall estimates are needed
(e.g., daily rainfall), the bottom up approach has the advantage to require a much lower number
of measurements and, hence, of satellite sensors. The limitations of the bottom up approach are
the possibility to estimate only terrestrial rainfall and its dependence on land characteristics
(e.g., low accuracy for dense vegetation coverage and complex topography, Brocca et al., 2014).
The bottom up approach has been applied over a range of scales: global (Crow et al.,
2011; Brocca et al., 2014; Ciabatta et al., 2018), continental (Wanders et al., 2015; Brocca et
al., 2016), and local (Massari et al., 2014; Brocca et al., 2015; Román-Cascón et al., 2017) scale.
Moreover, different satellite soil moisture products have been considered including SMOS (Soil
Moisture Ocean Salinity mission, Brocca et al., 2016), ASCAT (Advanced SCATterometer,
Brocca et al., 2017), AMSR-E (Advanced Microwave Scanning Radiometer, Crow et al., 2009),
and SMAP (Soil Moisture Active and Passive, Koster et al., 2016; Tarpanelli et al., 2017; Zhang
et al., 2019). First studies employing satellite rainfall estimates obtained through the bottom up
approach for hydrological and water resources applications have been recently published (e.g.,
Ciabatta et al., 2016; Abera et al., 2017; Brunetti et al., 2018; Camici et al., 2018). These studies
have highlighted the large potential of this technique as a complimentary and useful approach
for estimating rainfall from space, and have also shown its main limitations. Specifically, the



temporal resolution and the accuracy of satellite soil moisture products play a fundamental role
in determining the accuracy of the bottom up rainfall estimates.
In this study, we describe the newly developed SM2RAIN-ASCAT rainfall dataset
covering the period 2007-2018 and characterized by a spatial/temporal sampling of 12.5 km/1-
day. The new SM2RAIN-ASCAT dataset is obtained from the application of SM2RAIN
algorithm (Brocca et al., 2014) to ASCAT soil moisture data, and it is the first SM2RAIN-
ASCAT dataset available at the same spatial resolution as the ASCAT soil moisture dataset
(previous datasets have been under-sampled at 0.5- and 1-degree resolution). Moreover, we
have included the latest improvements in pre- and post-processing of soil moisture and rainfall
data as well as in the SM2RAIN algorithm. The main differences with the SM2RAIN-CCI
rainfall dataset (Ciabatta et al., 2018) are the input soil moisture dataset (the input of
SM2RAIN-CCI is the European Space Agency Climate Change Initiative Soil Moisture, ESA
CCI soil moisture, product, Dorigo et al., 2017), and the time coverage (SM2RAIN-CCI spans
the period 1998-2015). Technically, the use of the same satellite sensor in SM2RAIN-ASCAT
dataset is preferable to ensure consistency between soil moisture estimates over time to which
the SM2RAIN algorithm is highly sensitive.
The purpose of this study is twofold. As a first objective, we have applied SM2RAIN
algorithm at 1009 points uniformly distributed in the United States, Italy, India and Australia
for testing different configurations of data pre-/post-processing and SM2RAIN model equation.
This analysis has allowed us to select the best configuration that is implemented on a global
scale for obtaining the SM2RAIN-ASCAT dataset. The second objective is the assessment of
the global scale SM2RAIN-ASCAT dataset through the comparison with reference datasets and
by exploiting the Triple Collocation (TC) approach (Massari et al., 2017a). As reference
datasets we have used high-quality local raingauge networks from 2013 to 2017 in the United
States, Italy, India and Australia and three additional global datasets: the latest reanalysis of the
European Centre for Medium-Range Weather Forecasts (ECMWF), ERA5, the gauge-based
Global Precipitation Climatology Centre (GPCC), and the GPM IMERG product (Early Run
version).
**2  Datasets**
Nine different datasets have been collected for this study which are based on remote
sensing, ground observations and reanalysis. Refer to *Table 1* for a summary of the datasets.





The main input dataset for producing SM2RAIN-ASCAT dataset is the ASCAT soil
moisture data record provided by the "EUMETSAT Satellite Application Facility on Support
to Operational Hydrology and Water Management (H SAF)". ASCAT, currently on board
Metop-A (launched on October 2006), Metop-B (September 2012) and Metop-C (November
2018) satellites, is a scatterometer operating at C-band (5.255 GHz) and, by using the TU Wien
algorithm (Wagner et al., 2013) the H SAF provides a soil moisture product characterized by
12.5 km spatial sampling. The temporal sampling is varying as a function of latitude and the
number of satellites: by using Metop-A only a daily sampling is obtained, by using Metop-A
and Metop-B two observations per day are available at mid-latitudes. Here we have used the
H SAF ASCAT soil moisture data record (using Metop-A and Metop-B) available through the
product H113 (PUM, 2018) from 2007 to 2017 and its extension product H114 for the year
158    2018.

Three datasets obtained from the latest reanalysis of ECMWF, i.e., ERA5, have been
used. ERA5 reanalysis is characterized by a spatial resolution of ~36 km and hourly temporal
resolution. ERA5 is available from the Copernicus Climate Change service and the datasets
cover the period 1979 to present. We have extracted hourly observations for the period 2007-
2018 for three variables: evaporation, soil temperature for the first layer (0-7 cm) and total
rainfall (computed as the difference between total precipitation and snowfall). Evaporation data
are used as additional input to the SM2RAIN algorithm and soil temperature data for masking
periods with frozen soils. Total rainfall has been considered as a benchmark for the calibration
of global SM2RAIN parameter values (see next section).
Ground-based rainfall datasets from regional networks have been also collected including
the Climate Prediction Center (CPC) Unified Gauge-Based Analysis of Daily Precipitation in
the United States, the gridded rainfall data provided by ~3000 stations of the National
Department of Civil Protection in Italy (Ciabatta et al., 2017), the India Meteorological
Department (IMD, http://www.imd.gov.in/pages/services_hydromet.php) rainfall observations
in India, and the Australia Water Availability Project (AWAP,
http://www.bom.gov.au/jsp/awap/rain/index.jsp) gridded rainfall data in Australia. These
datasets have been used firstly for the selection of the optimal configuration of SM2RAIN
implementation. For that, 1009 points uniformly distributed over the four regions have been
selected. Secondly, the regional networks have been used for the assessment of the global
SM2RAIN-ASCAT rainfall product at regional scale.



The ERA5 and local rainfall datasets have been regridded over the ASCAT grid (12.5 km)
through the nearest neighbouring method and resampled at daily time scale as accumulated
rainfall from 00:00 to 23:59 UTC. The ERA5 evaporation and soil temperature data are also
regridded to the same grid and aggregated at daily scale as accumulated and average value from
00:00 to 23:59 UTC, respectively.
For the global assessment of SM2RAIN-ASCAT, two additional rainfall datasets have
been considered: Global Precipitation Climatology Centre (GPCC) Full Data Daily Product
(Schamm et al., 2015) and GPM IMERG Early Run product (Hou et al., 2014), hereinafter
referred to as GPM-ER. Due to the availability of GPM-ER from April 2014, the global analysis
has been carried out in the 4-year period from January 2014 to December 2018. Moreover, for
the global inter-comparison all the datasets (SM2RAIN-ASCAT, ERA5, GPCC, and IMERG-
ER) have been regridded at 0.25-degree resolution by spatial averaging the pixels contained in
each 0.25-degree cell for SM2RAIN-ASCAT and GPM-ER, and by selecting the nearest pixel
for ERA5 and GPCC.
**3    Methods**
In the following, the methodology used for obtaining the SM2RAIN-ASCAT dataset is
described. Specifically, three steps are carried out (see ***Figure 1***): 1) surface soil moisture data
pre-processing, 2) SM2RAIN algorithm, and 3) rainfall data post-processing.
**3.1    Soil moisture data pre-processing**
The ASCAT surface soil moisture product is provided as relative soil moisture (between
0 and 1) at the overpass time of the satellite sensor. For the application of SM2RAIN algorithm,
data should be equally spaced in time and hence, we have linearly interpolated in time soil
moisture observations every 24 hours, 12 hours and 8 hours. In a preliminary test (not shown
for brevity), we have tested the three sampling frequencies with the baseline formulation for
SM2RAIN (***equation 6***, see below). The best performances have been obtained with 12 hours
sampling, particularly from 2013 to 2018 in which both Metop-A and -B are available.
Therefore, 12 hours sampling has been used in the following analyses.
One of the major problems in using satellite soil moisture observations for rainfall
estimation is related to the high frequency fluctuations caused by measurement and retrieval
errors. If positive, such fluctuations are interpreted erroneously as rainfall by SM2RAIN



algorithm. Therefore, satellite surface soil moisture data need to be filtered before being used
as input into SM2RAIN. In previous studies, the exponential filtering has been considered
(Wagner et al., 1999). The exponential filter, also known as Soil Water Index (SWI), has been
used for filtering surface soil moisture time series as a function of a single parameter, *T*, i.e.,
the characteristic time length. In this study, we have tested two additional filtering methods.
The first one is an extension of the exponential filter in which the *T* parameter is assumed to be
varying with soil moisture as proposed in Brocca et al. (2013b). Specifically, *T* decreases with
increasing soil moisture through a 2-parameter power law. Therefore, the data are filtered more
during dry conditions. The third approach that we have implemented is a discrete wavelet filter
(similar to Massari et al., 2017b). The discrete wavelet filter cuts the higher frequencies of the
signal, typically characterized by noises, over a threshold selected through the principle of
Stein's Unbiased Risk at multiple levels. We have found the Daubechies wavelets to be the most
appropriate functions because their shape and the shape of the soil moisture signal is similar.
Therefore, we have implemented a Daubechies-based wavelet filter in which the filtering level
is optimized.
For all the filtering approaches, the parameter values of the filters have been optimized
point-by-point in order to reproduce the reference rainfall observations.

## 3.2   SM2RAIN algorithm and calibration

The SM2RAIN algorithm is based on the inversion of the soil water balance equation and
allows to estimate the amount of water entering the soil by using as input soil moisture
observations from in situ or satellite sensors (e.g., Brocca et al., 2013a; 2014; 2015; Koster et
al., 2016; Ciabatta et al., 2017; Massari et al., 2014). Specifically, the soil water balance
equation can be described by the following equation (over non-irrigated areas):
$$nZ\frac{dS(t)}{dt} = p(t) - g(t) - sr(t) - e(t) \qquad (1)$$
where *n* [-] is the soil porosity, *Z* [mm] is the soil layer depth, *S(t)* [-] is the relative
saturation of the soil or relative soil moisture, *t* [days] is the time, *p(t)* [mm/day] is the rainfall
rate, *g(t)* [mm/day] is the drainage (deep percolation plus subsurface runoff) rate, *sr(t)*
[mm/day] is the surface runoff rate and *e(t)* [mm/day] is the actual evapotranspiration rate.
For estimating the rainfall rate, **equation (1)** is applied only during rainfall periods and,
hence, some of the components of the equation can be considered as negligible. For instance,





the actual evapotranspiration rate during rainfall is quite low due to the presence of clouds and,
hence, the absence of solar radiation. Similarly, the surface runoff rate, i.e., the water that does
not infiltrate into the soil and flows at the surface to the watercourses, is much lower than the
rainfall rate, mainly if **equation (1)** is applied at coarse spatial resolution (20 km), i.e., with
satellite soil moisture data. Indeed, most of water becomes runoff flowing in the subsurface,
and also the part that does not infiltrate, due to almost impervious land cover or soil, may re-
infiltrate downstream within a pixel at 20 km scale.
Following the indications obtained in Brocca et al. (2015), we have assumed the surface
runoff rate, *sr(t)*, as negligible (i.e., Dunnian runoff) and we have rearranged **equation (1)** for
estimating the rainfall rate:
$$p(t) = nZ\frac{dS(t)}{dt} - g(t) - e(t) \tag{2}$$

In this study, we have considered different formulations for equation (2) by varying the
drainage rate as:
$$g(t) = K_s S(t)^m \tag{3.1}$$

$$g(t) = K_s S(t)^{\lambda+1}\left[1 - \left(1 - S(t)^{\frac{\lambda+1}{\lambda}}\right)^{\frac{\lambda}{\lambda+1}}\right]^2 \tag{3.2}$$

$$g(t) = K_s S(t)^{\tau}\left[1 - \left(1 - S(t)^{\frac{1}{m}}\right)^{m}\right]^2 \tag{3.3}$$

where $K_s$ [mm/day] is the saturated hydraulic conductivity, $m$ [-] and $\lambda$ [-] are exponents related
to the pore size distribution index, and $\tau$ is the tortuosity index. Specifically, the three equations
represent the hydraulic conductivity - soil moisture formulation by Brooks-Corey (3.1), van
Genuchten (3.2), and Mualem-van Genuchten (3.3).
The actual evapotranspiration rate has been considered as an additional input, together
with soil moisture, here obtained from ECMWF reanalysis ERA5:
$$e(t) = K_c\, ET_{ERA5}(t) \tag{4}$$

where $ET_{ERA5}(t)$ [mm/day] is the actual evapotranspiration rate obtained from ERA5 reanalysis
and $K_c$ [-] is a correction factor for taking into account potential bias in ERA5 estimates.



Moreover, we have considered an additional formulation in which $Z$ is a function of soil
moisture taking into account the different penetration depth of satellite sensors as a function of
wetness conditions:
$$Z = Z[0.1 + (1 - S(t)^c)] \qquad (5)$$
where $c$ exponent determines the rate of decrease of penetration depth with increasing soil
moisture.
Accordingly, we have used different formulations for equation (2) that are compared with
the baseline equation used in previous studies (e.g., Brocca et al., 2014):
$$p(t) = Zn\frac{dS(t)}{dt} + K_s S(t)^m \qquad (6)$$
In synthesis, we have investigated 3 different configurations (total of 5 options) for: 1)
selecting the best equation for the drainage rate (*equations 3*), 2) testing the possibility to
include the evapotranspiration component (*equation 4*), and 3) testing the use of a variable
penetration depth with soil moisture conditions (*equation 5*). Each new configuration has been
compared with the baseline (*equation 6*) in order to select the best configuration for SM2RAIN
algorithm (see *Figure 1*). For all configurations, negative rainfall values, that might occur
during some dry-down cycles, have been set equal to zero.
**3.3   Rainfall data post-processing**
The use of satellite soil moisture observations for obtaining rainfall estimates is affected
by errors in the input data and in the retrieval algorithm SM2RAIN. The correction of the
overall bias in the climatology is a simple and effective approach for mitigating part of such
errors. Specifically, we refer here to a static correction procedure that once calibrated for a time
period can be applied in the future periods, also for real-time products. We have implemented
two different approaches for climatological correction: 1) a cumulative density function (CDF)
matching approach at daily time scale, and 2) a monthly correction approach. Specifically, the
implemented CDF matching approach is a 5-order polynomial correction as described in Brocca
et al. (2011) for matching the CDF of estimated rainfall with respect to reference rainfall, in
which the CDF are computed over the whole calibration period at daily time scale. The monthly
correction approach computes the monthly ratios between the climatology of estimated and
reference rainfall, i.e., 12 correction factors per pixel. Then, the SM2RAIN-estimated rainfall



is multiplied for the monthly correction factors to obtain the climatologically corrected
SM2RAIN-estimated rainfall.

### 3.4   Triple collocation analysis

For the global assessment of satellite, reanalysis and gauge-based rainfall products we

have used the Triple Collocation (TC) technique. TC can theoretically provide error and
correlations of three products (a triplet) given that each of the three products is afflicted by
mutually independent errors. Therefore, in principle, TC can be used for assessing the quality
of satellite products without using ground observations (Massari et al., 2017a). In this study,
we have implemented the same procedure as described in Massari et al. (2017) and we refer the
reader to this study for the analytical details. In synthesis, by using the extended TC method
firstly proposed by McColl et al. (2014), it is possible to estimate the temporal correlation, $R_{TC}$,
of each rainfall product in the triplets with the truth.

### 3.5   Performance scores

Several metrics have been used to assess the product performance during the validation

period. As continuous scores we have computed the Pearson's correlation coefficient (R), the
root mean square error (RMSE), the mean error between estimated and reference rainfall
(BIAS), and the ratio of temporal variability of estimated and reference rainfall (STDRATIO).
Continuous scores have been computed on a pixel-by-pixel basis by considering 1 day of
accumulated rainfall. Moreover, three categorical scores, i.e. Probability of Detection (POD),
False Alarm Ration (FAR) and Threat Score (TS), have been computed. POD is the fraction of
correctly identified rainfall events (optimal value POD=1), FAR is the fraction of predicted
events that are non-events (optimal value FAR=1), while TS provides a combination of the
other two scores (optimal value TS=1). The categorical assessment is carried out by considering
a rainfall threshold of 0.5 mm/day (instead of 0 mm/day) in order to exclude spurious events
that might be due to rainfall interpolation\regridding in the reference datasets. For a complete
description of the categorical scores the reader is referred to Brocca et al. (2014).

### 4   Results

The results are split in three parts: 1) selection of the optimal configuration of SM2RAIN

through the assessment at 1009 points, 2) generation of global SM2RAIN-ASCAT rainfall



dataset, and 3) regional assessment of SM2RAIN-ASCAT with gauge-based rainfall datasets
and global assessment through TC.

### 4.1 Selection of the best SM2RAIN processing configuration at 1009 points

As a first step we have co-located satellite soil moisture data from ASCAT, ground-based
rainfall observations and actual evapotranspiration data from ERA5 in space and time at 1009
points. We have selected 1009 points uniformly distributed, instead of the whole region, in
order to test multiple configuration at reasonable computational cost. These datasets are made
freely available here (https://doi.org/10.5281/zenodo.2580285, Brocca, 2019) for those
interested to test alternative approaches for rainfall estimation from ASCAT soil moisture.
Specifically, we have considered the period 2013-2016, 2013-2014 for the calibration and
2015-2016 for the validation; in the sequel only the results in the validation period are shown.
The reference configuration, REF, as used in previous SM2RAIN applications (e.g., Brocca et
al., 2014), uses the SWI for data filtering, the SM2RAIN formulation as in *equation (6)*, and
no climatological correction. Results in the validation period are shown in *Figure 2A* in terms
of temporal R against reference data. As shown, the median R for all points is equal to 0.60,
with better results in Italy (median R=0.67, see boxplots) and similar results in the other 3
regions (median R=0.60 and 0.59). These results are in line with previous studies (e.g., Ciabatta
et al., 2017; Tarpanelli et al., 2017) carried out in Italy and India and highlight the potential of
ASCAT soil moisture observations to provide daily rainfall estimates. *Figure 3* (first column)
shows the results for the different performance metrics; in the last column the results obtained
with GPM-ER are shown for comparison. Very good statistics have been obtained in terms of
RMSE and BIAS but a tendency to underestimate the observed rainfall variability (median
STDRATIO=0.60) and medium-high probability of false alarm (median FAR=0.53). The other
scores are similar, or slightly lower than those obtained through GPM-ER.
The first test has been dedicated to the filtering of soil moisture data by using three
approaches: 1) SWI, i.e., the REF configuration, 2) SWI with T varying with soil moisture,
SWI-Tvar, and 3) the discrete wavelet filtering, WAV. *Figure 3* shows in the first three columns
the summary of the performance scores highlighting that the SWI-Tvar configuration is
performing the best, even though the differences with REF configuration are small. *Figure 2b*
shows the R map for SWI-Tvar configuration while in *Figure 2c* the differences in R-values
with REF are displayed. Improved performance in terms of R is visible over most of the pixels
except in central Australia.





The second test has been performed on the SM2RAIN equation by using different

drainage functions (VGEN and MUA configurations), by adding the evapotranspiration
component (EVAP), and by considering the variability of sensing depth, $Z$, with soil moisture
(ZVAR); the details of the different configurations are given in *Table 2*. VGEN, MUA and
ZVAR configurations are characterized by lower performances than REF (see *Figure 3*,
columns 4, 5 and 7), even though MUA and ZVAR incorporate an additional parameter to be
calibrated (and, hence, better performance was expected). The addition of evapotranspiration
brings a slight improvement with respect to REF (see *Figure 3*, columns 6), with results similar
to SWI-Tvar. Larger improvements are obtained over areas in which evapotranspiration is more
important, e.g., in the south-western United States and central western Australia. In India and
Italy, the results are very similar to REF. However, EVAP configuration requires actual
evapotranspiration data from ERA5 as additional input and such data might be not available for
the implementation of the processing algorithm in an operational context.

The final test has been done by applying the daily CDF matching, BC-CDF, and monthly

correction factors, BC-MON, for correcting the climatological bias in SM2RAIN-derived
rainfall estimates; results are shown in columns 8 and 9 of *Figure 3*. For these two
configurations, the improvements with respect to REF are evident but with different magnitude
for the different scores. BC-CDF improves significantly STDRATIO, TS and FAR with a slight
deterioration in R and RMSE. BC-MON shows the highest R-values among all configurations
with the larger improvements in India, Italy and United States. However, the improvement in
terms of STDRATIO, TS and FAR is less important than BC-CDF. Therefore, depending on
which score is assumed more important, one of the two configurations can be selected. If
compared with GPM-ER, BC-CDF and BC-MON configurations show similar results with
larger positive differences, in terms of RMSE, BIAS, STDRATIO and POD; R values are
slightly better for GPM-ER that is also much better in terms of TS and FAR.

*Figure 4* shows time series of rainfall averaged over the four regions as obtained from

ground observations and from BC-MON configuration. The agreement of spatially averaged
rainfall with observations is high with R-values greater than 0.83, very low BIAS and
KGE>0.67. Moreover, regional scale rainfall events are correctly reproduced both in terms of
timing and magnitude.





## 4.2 Generation of SM2RAIN-ASCAT dataset


Based on the tests performed in the previous paragraph, we have selected the best
configuration using SWI-Tvar for filtering, Brooks-Corey function for losses, and the monthly
correction approach for climatological correction. The addition of evapotranspiration
component, even though showing some improvements, has been not used in view of an
operational implementation of the method. The monthly correction approach has been selected
as R and RMSE scores have been considered more important.
The selected configuration has been applied on a global scale to 839826 points over which
ASCAT soil moisture observations are available. As reference dataset for calibrating the
filtering, SM2RAIN, and post-processing parameter values, the ERA5 rainfall has been used
mainly because of its higher spatial resolution compared to GPCC (36 km versus 100 km).
However, we have also tested the use of the two datasets for calibration at randomly chosen
20000 points which showed that the estimated rainfall in the two calibration tests is very similar.
For instance, the median R between the two SM2RAIN-ASCAT dataset is higher than 0.90 (not
shown for brevity). This result clearly demonstrate that the selection of reference dataset has a
small influence on SM2RAIN-derived rainfall that is mostly driven from soil moisture temporal
fluctuations. Additionally, considering the improved ASCAT coverage after 2013, the
calibration has been split from 2007 to 2012 (Metop-A) and from 2013 to 2018 (Metop-A and
-B). Finally, being aware of the regions in which ASCAT soil moisture product is performing
not good, we have flagged rainfall observations in space and time when the data are supposed
to be less reliable. In space, we have used the committed area mask developed for ASCAT soil
moisture product (PVR 2017) in which the quality of ASCAT soil moisture is found to be good,
a frozen probability mask and a topographic complexity mask. In time, we have considered the
soil temperature data from ERA5 and flagged the observations with soil temperature values
between 0°C and 3°C as temporary frozen soil and below 3°C as frozen soil. As many
applications require continuous data, we have preferred to flag the data instead of removing
them with an expected loss of accuracy.
The SM2RAIN-ASCAT dataset so obtained has a spatial sampling of 12.5 km, a daily
temporal resolution and covers the 12-year period 2007-2018. *Figure 5* shows R and RMSE
values between SM2RAIN-ASCAT and ERA5 in a single map. Green light colours indicate
very good performance with high R and low RMSE, orange to red colours indicate low R and
high RMSE, while black indicates low RMSE and R highlighting areas in which rainfall





occurrence and variability is very low (e.g., Sahara Desert, high latitudes). The dataset has been
found to perform accurately (high R and low RMSE) in western United States, Brazil, southern
and western South America, southern Africa, Sahel, southern-central Eurasia, and Australia.
The areas in which SM2RAIN-ASCAT is characterized by lower accuracy are those with dense
vegetation (Amazon, Congo, and Indonesia), with complex topography (e.g., Alps, Himalaya,
Andes), at high latitudes and Saharan and Arabian deserts. In these areas it is well-known that
ASCAT soil moisture product has limitations (e.g., Wagner et al., 2013), and generally the
retrieval of soil moisture from remote sensing is more challenging. The median R and RMSE
values are equal to 0.56 and 3.06 mm/day, with slightly better scores in the period 2013-2018
(R=0.57, RMSE=2.95), thanks to the availability of ASCAT on both Metop-A and -B.
**4.3   Regional and global assessment of SM2RAIN-ASCAT dataset**

By using all the pixels included in the four regions (Italy, United States, India and

Australia), for a total of 29843 points, the new SM2RAIN-ASCAT rainfall dataset has been
compared with reference rainfall observations in *Figure 6*, by considering the whole period
2007-2018. Specifically, the box plots of R and RMSE scores are shown and also compared
with the results obtained through GPCC, ERA5, and GPM-ER. In terms of R, SM2RAIN-
ASCAT show better performance in Italy (median R=0.67) and United States (median R=0.62),
almost comparable with the other datasets; in Australia and India R-values are lower (median
R=0.61 and 0.59). In the selected regions, the best product is GPCC (mainly in Australia)
followed by ERA5 while GPM-ER and SM2RAIN-ASCAT performing similarly. The better
performance of GPCC are expected (gauge-based dataset) and also the very good performance
of ERA5 in Italy and Australia thanks to the availability of ground observations for the
reanalysis. When considering the RMSE score, the results are quite different with respect to R.
SM2RAIN-ASCAT is overall very good being the best (second best) product in United States
(India). The ranking of the product is GPCC, SM2RAIN-ASCAT, ERA5 and GPM-ER, with
the latter showing high RMSE values in United States and Australia. As obtained in previous
studies (Brocca et al., 2016; Ciabatta et al., 2017), the SM2RAIN approach is very good in
obtaining low RMSE values thanks to its accuracy in the retrieval of accumulated rainfall.
Moreover, the product accuracy is stable over time as it is not as strongly affected by the
availability of satellite overpasses as in the top down approach.

On a global scale, the TC approach has been implemented by using the triplet SM2RAIN-

ASCAT, GPM-ER and GPCC, by considering the common period 2015-2018. Theoretically,



the extended TC approach provides the correlation, $R_{TC}$, against the underlying "truth". **Figures**
**7A and 7B** show the $R_{TC}$ maps for SM2RAIN-ASCAT and GPM-ER highlighting similar mean
values (0.66 and 0.64 for SM2RAIN-ASCAT and GPM-ER, respectively). It should be
underlined that the $R_{TC}$ values are higher than those obtained in the comparison with ground
observations as theoretically the metric does not contain the error in the reference (Massari et
al., 2017a). The spatial pattern of the performance is quite different as demonstrated in **Figure**
**7c** in which the differences between the two $R_{TC}$ maps is shown. Again, these results underline
the strong complementarity between bottom up and top down approaches (e.g., Ciabatta et al.,
2017; Chiaravallotti et al., 2018). As expected, SM2RAIN-ASCAT performs worse over desert
areas, tropical forests and complex mountainous regions. Differently, over plains and low
vegetated areas SM2RAIN-ASCAT is performing better than GPM-ER, particularly in the
southern hemisphere. Indeed, in Africa and South America SM2RAIN-ASCAT provides good
performance (see also **Figure 7A**) thanks to the capability of the bottom up approach to estimate
accumulated rainfall accurately with a limited number of satellite overpasses occurring in these
areas, differently from the top down approach used in GPM-ER.
The box plots of $R_{TC}$ shown in **Figure 7D** indicates that, overall, GPCC is performing
similar to the two satellite products with major differences in the spatial patterns of the
performance. SM2RAIN-ASCAT is performing the best among the three products in Africa,
South America, central-western United States and central Asia while GPCC is performing the
best in the remaining parts except the tropical region in which GPM-ER is performing very
good (see **Figure 8**). If we consider only the committed area of ASCAT (PVR 2017), in which
the soil moisture product is supposed to be reliable, the mean value of $R_{TC}$ is equal to 0.72
whereas in the masked area it is equal to 0.59.
**5   Data availability**
The         SM2RAIN-ASCAT         dataset         is         freely         available         at
https://doi.org/10.5281/zenodo.2591215 (Brocca et al., 2019).
**6   Conclusions**
In this study, we have described the development of a new SM2RAIN-ASCAT rainfall
dataset highlighting the steps carried out for improving the retrieval algorithm and the pre-/post-
processing of the data. The major novelties of the SM2RAIN-ASCAT rainfall dataset



developed here with respect to previous versions are: 1) application of SM2RAIN at full spatial
resolution thus providing a gridded dataset with sampling of 12.5 km, 2) improved sampling
and filtering of ASCAT soil moisture data, 3) application of monthly climatological correction.
The SM2RAIN-ASCAT dataset has been preliminary assessed at regional (***Figures 4 and***
***6***) and global (***Figure 5, 7 and 8***) scale in terms of different performance metrics with larger
emphasis on the temporal correlation, R, and the root mean square error, RMSE. The overall
performances are good, mainly in terms of RMSE thanks to the capacity of SM2RAIN to
accurately reproduce accumulated rainfall consistently over time. Importantly, SM2RAIN-
ASCAT is found to perform even better than ground-based GPCC product over the southern
hemisphere in Africa and South America, and also in central-western United States and central
Asia.
The SM2RAIN-ASCAT rainfall dataset can now be used as input for applications such
as flood prediction (similarly to Camici et al., 2018 and Massari et al., 2018), landslide
prediction (Brunetti et al., 2018) and novel applications for the agriculture and for the water
resources management.

**Acknowledgments:** The authors gratefully acknowledge support from the EUMETSAT
through the Global SM2RAIN project (contract n° EUM/CO/17/4600001981/BBo) and the
"Satellite Application Facility on Support to Operational Hydrology and Water Management
(H SAF)" CDOP 3 (EUM/C/85/16/DOC/15).



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



**Tables**
**Table 1.** List of satellite, ground-based and reanalysis products used in this study (the
spatial/temporal sampling used in this study is reported).

| Short name | Full name and details | Data source | Spatial/ temporal sampling | Time period | References |
|---|---|---|---|---|---|
| **SOIL MOISTURE** | | | | | |
| ASCAT | Advanced Scatterometer | satellite | 12.5 km/ daily | 2007 - present | Wagner et al. (2013) |
| **RAINFALL** | | | | | |
| ERA5 | ECMWF | reanalysis | 0.25°/ daily | 1979 - present | https://cds.climate.copernicus.eu/cdsapp#!/dataset/reanalysis-era5-single-levels?tab=overview |
| GPCC | Global Precipitation Climatology Centre Full Data Reanalysis | gauge | 1°/ daily | 1988 - present | Schamm et al. (2015) |
| IMERG Early Run | Global Precipitation Measurement | satellite | 0.1°/ daily | 2014 - present | Hou et al. (2014) |
| CPC | Climate Prediction Center – United States | gauge | 0.5°/ daily | 1948 - present | https://www.esrl.noaa.gov/psd/data/gridded/data.unified.daily.conus.html |
| ITA-DPC | Gauge-based rainfall dataset –Italy | gauge | 0.1°/ daily | 2008 - present | Ciabatta et al. (2017) |
| AWAP | Australian Water Availability Project – Australia | gauge | 0.05°/ daily | 1900 - present | http://www.bom.gov.au/jsp/awap/rain/index.jsp |
| IMD | India Meteorological Department - India | gauge | 0.25°/ daily | 1901 - present | http://www.imd.gov.in/pages/services_hydromet.php |
| **SOIL TEMPERATURE and EVAPOTRANSPIRATION** | | | | | |
| ERA5 | ECMWF | reanalysis | 0.25°/ daily | 1979 - present | https://cds.climate.copernicus.eu/cdsapp#!/dataset/reanalysis-era5-single-levels?tab=overview |




**Table 2.** Configurations used in the paper (SWI: Soil Water Index, BCO: Brooks-Corey, VGE:
van Genuchten, MUA: Mualem-van Genuchten, SWI-Tvar: SWI with T varying with soil
moisture, WAV: wavelet filtering, CDF: climatological correction with daily cumulative
density function matching, MON: climatological correction with monthly correction factors).

| Short Name | Filtering | Losses | Evapotranspiration | Depth varying | Climatological Correction |
|---|---|---|---|---|---|
| REF | SWI | BCO | no | no | no |
| SWI-Tvar | SWI-Tvar | BCO | no | no | no |
| WAV | WAV | BCO | no | no | no |
| VGEN | SWI | VGE | no | no | no |
| MUA | MUA | VGE | no | no | no |
| EVAP | SWI | BCO | yes | no | no |
| ZVAR | SWI | BCO | no | yes | no |
| BC-CDF | SWI-Tvar | BCO | no | no | CDF |
| BC-MON | SWI-Tvar | BCO | no | no | MON |




**Figures**

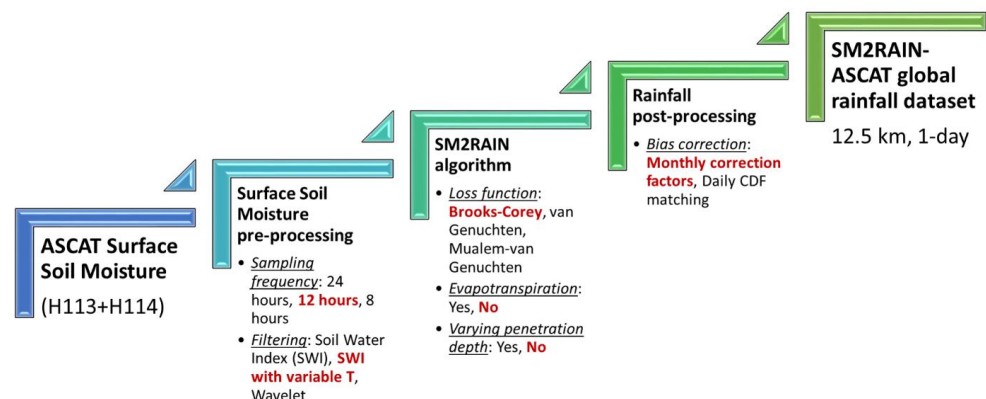


**Figure 1.** Processing steps for obtaining the SM2RAIN-ASCAT global rainfall dataset (2007-
2018) from ASCAT surface soil moisture data: pre-processing, SM2RAIN algorithm, and post-
processing. Each bullet represents a possible configuration that has been tested, the selected
configuration is in red, bold font.

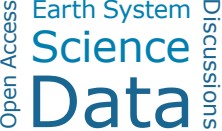



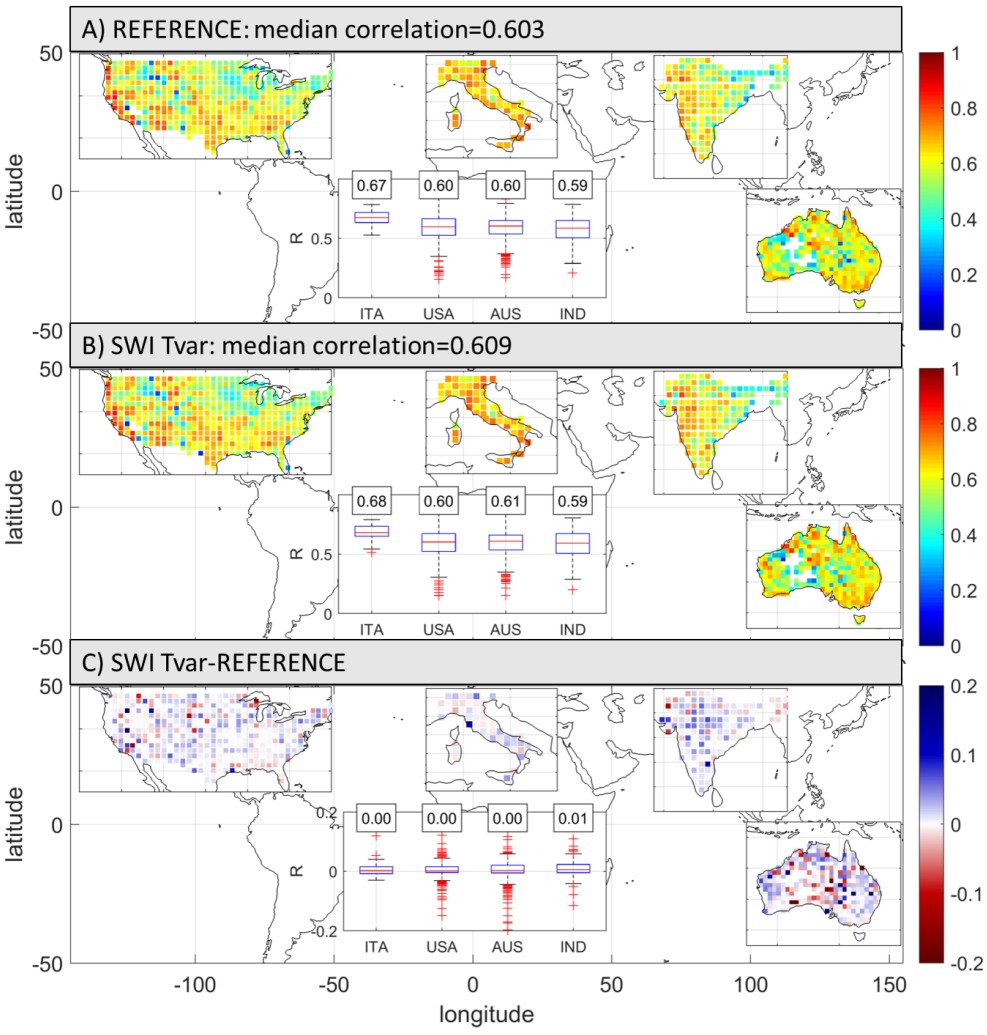


**Figure 2.** Performance of two different configurations at 1009 points in terms of Pearson's
correlation, R [-]. A) R map with reference configuration, B) R map with Soil Water Index
(SWI) filtering with variable T as a function of soil moisture, and C) R map difference (B)-(A).
The inner box plots show the R values (and R differences) split for different regions.

677





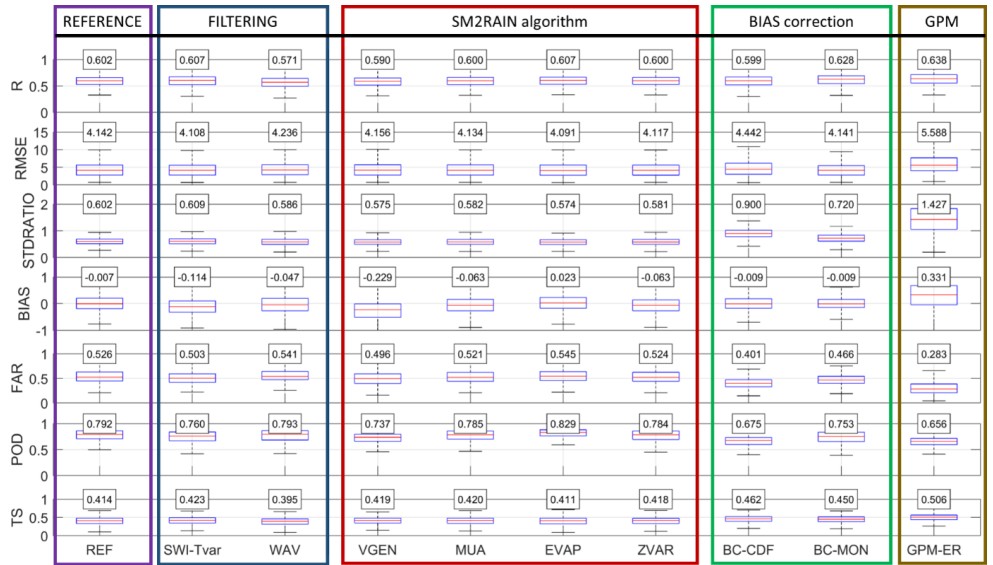

678

**Figure 3.** Performances at 1009 points in terms of Pearson's correlation, R [-], root mean square
error, RMSE [mm/day], variability ratio, STDRATIO [-], BIAS [mm/day], false alarm ratio,
FAR [-], Probability of Detection, POD [-], and Threat Score, TS [-]. For details of the different
configurations see Table 2 (GPM-ER: GPM IMERG Early Run product).




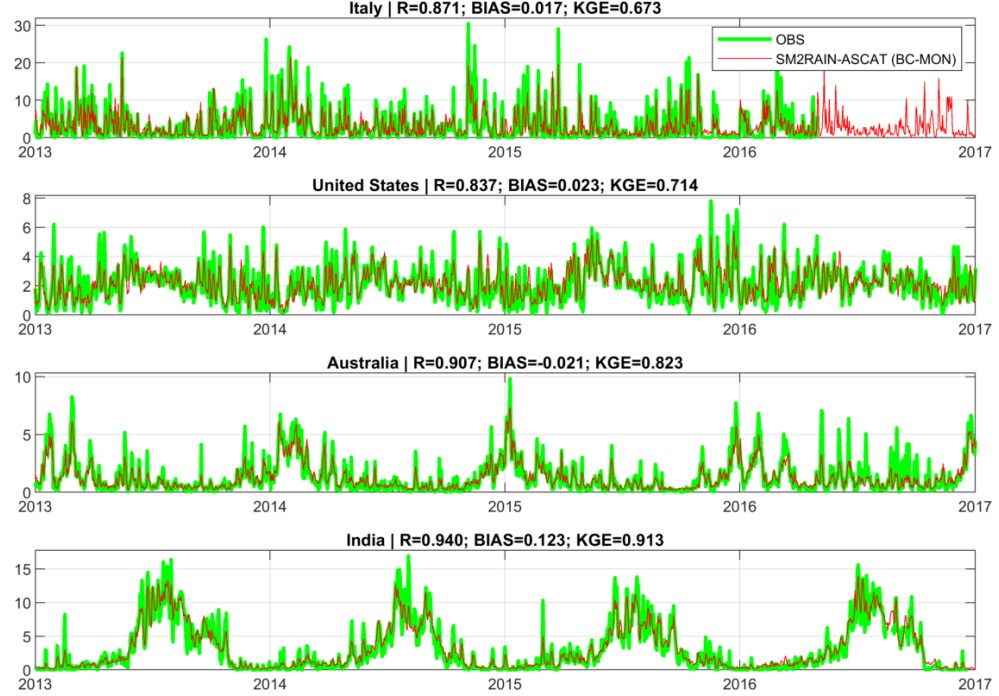


**Figure 4.** Time series of mean areal rainfall for the four regions for observed data, OBS, and
SM2RAIN-ASCAT dataset, BC-MON configuration (R [-]: Pearson's correlation, BIAS
[mm/day]: mean error, KGE [-]: Kling-Gupta efficiency).



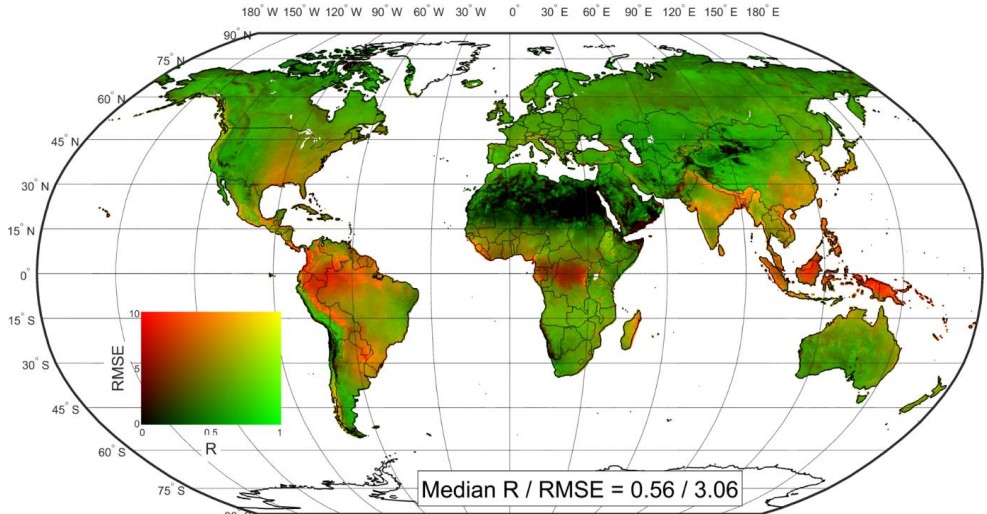


**Figure 5.** Pearson's correlation, R, and root mean square error, RMSE, map of SM2RAIN-ASCAT dataset compared with ERA5 reanalysis dataset used as benchmark (period 2007-2018). The analysis is carried out at 1-day and 12.5 km temporal and spatial resolution. The map shows that SM2RAIN-ASCAT dataset is performing well in the western United States, Brazil, southern and western South America, southern Africa, Sahel, southern-central Eurasia, and Australia (green colours).

696

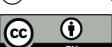



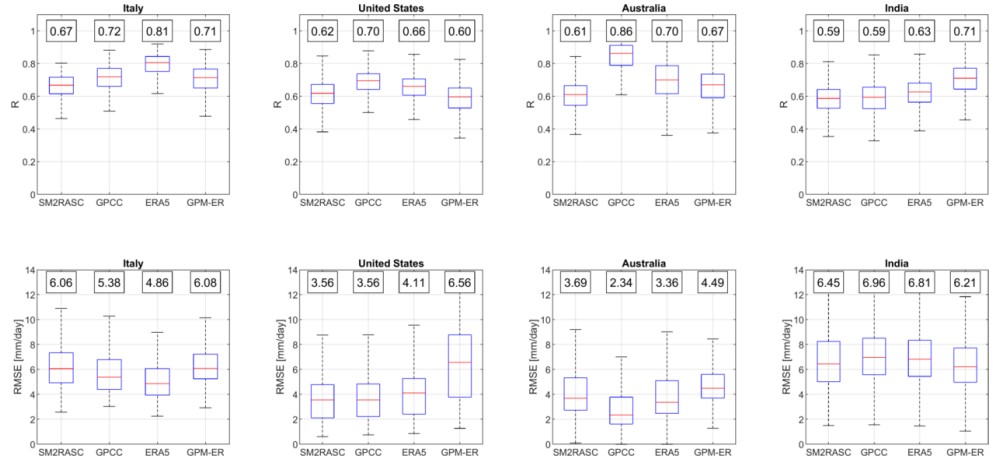

697

**Figure 6.** Regional assessment of SM2RAIN-ASCAT rainfall dataset and comparison with
GPCC, ERA5 and GPM-ER rainfall products. As reference the high-quality ground-based
datasets in Italy, United States, India and Australia are used. Top panels show the Pearson's
correlation, R, and the bottom panels the root mean square error, RMSE.

702



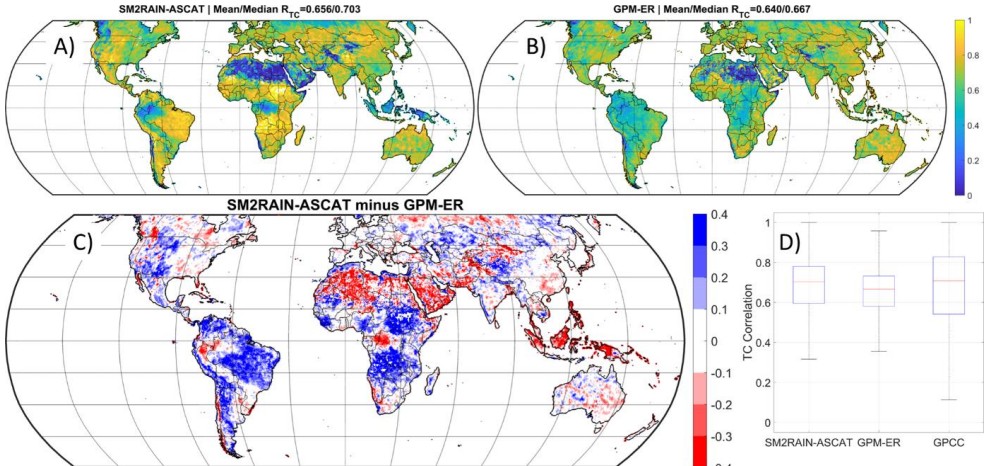

703

**Figure 7.** Global triple collocation, TC, results. A) $R_{TC}$ map for SM2RAIN-ASCAT, B) $R_{TC}$ map for GPM-ER, (C) differences between (A) and (B), i.e., blue (red) colours for pixels in which SM2RAIN-ASCAT (GPM-ER) is performing better, and D) box plot of $R_{TC}$ for SM2RAIN-ASCAT, GPM-ER, and GPCC. SM2RAIN-ASCAT is performing significantly better than GPM-ER in South America and Africa (excluding desert and tropical forest areas), elsewhere the two satellite products perform similarly.





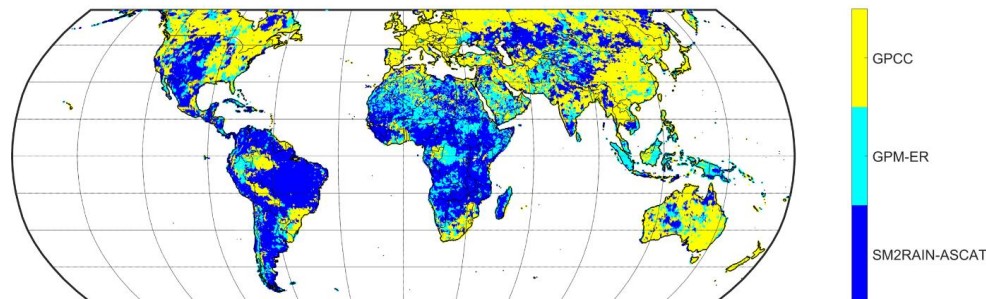


**Figure 8.** Best performing product based on the results of triple collocation shown in Figure 7.
SM2RAIN-ASCAT is performing the best among the three products in Africa, South America,
central-western United States and central Asia while GPCC is performing the best in the
remaining parts of the northern hemisphere and in Australia. GPM-ER is the best product in the
tropical and equatorial region.