# Peer review of "SM2RAIN-ASCAT (2007-2018): global daily satellite rainfall"

_Earth System Science Data, 2019_

## Referee Comment (RC1) · Anonymous Referee #1 · 22 May 2019

This study provides detailed descriptions of the SM2Rain product and several evaluation results. Overall, the study would be useful for current and future SM2Rain users, and fits the scope of ESSD. However, I do find that the manuscript misses several key information in SM2Rain production and evaluation.

1. Line 24 – 29: The statement here is too strong. I agree that SM2Rain is a useful product in some aspects. However, I have not seen strong evidences that SM2Rain substantially outperforms other merged products, e.g., MSWEP v2.0. Additionally, soil moisture retrievals prior 2002 have very low data quality. I personally doubt if good precipitation can be derived from these soil moisture data sets. Hence, I also suspect whether SM2Rain "is suited to build long-term consistent rainfall".

2. Line 147: A global map of ASCAT temporal sampling frequency would be helpful.

[Figure]

3. Line 201 – 203: I'm wondering if there are any risks of increasing false rainfall events by linear interpolation?

4. Line 242: The authors state that runoff at 20km grid is negligible. Can you provide some rainfall-runoff simulation works to support this hypothesis?

5. Line 249: I'm a little bit confused by equation 2. First, the authored stated that ET is negligible. Then, why it is still considered in equation 2? Second, it seems g(t) and e(t) should be plus in sign, according to equation (1)?

6. Line 261: e(t) is calculated using ERA5 ET. The ERA5 ET is expected to depend on ERA5 precipitation. For instance, a dry period seen by ERA5 (precipitation deficiency) will lead to low ET. Therefore, the authors should discuss the dependency of ERA5 and SM2Rain rainfall product, particularly when TC is considered in the later part of the paper.

7. Line 281: What is the reference rainfall?

8. Section 3.4: Please also specify the error model used in this TC analysis.

9. Section 3.5: Equations of these scores will be helpful here.

10. Line 367: I'm still not clear which product is used as a reference to correct SM2Rain.

11. Section 4.1 and Line 385: It's un clear how SM2Rain parameters were calibrated (determined) and extended to the global scale.

12. Figure 5: SM2Rain is calibrated against ERA5. Therefore, the consistency of ERA5 and SM2Rain only suggests how well ASCAT was fitted to ERA5. The authors should be clear that this is not suggesting the accuracy or the performance of SM2Rain (Line 414 – 415).

13. Line 431 – 433: SM2Rain show better performances relative to which product? It seems that SM2Rain's R is much lower than the other three in Figure 6 a and b.

14. Following the comment above, SM2Rain was derived by calibrations against ERA5. However, its performances are consistently lower than ERA5. Then, what's the contribution/value of SM2Rain?

15. Line 446: What products are used for TC analysis? Massari used ERA. However, I don't think this is appropriate for this study. SM2Rain here is calibrated against ERA, and they may have cross-correlated errors.

---

## Referee Comment (RC2) · Anonymous Referee #2 · 24 May 2019

An update of a satellite soil moisture-based rainfall dataset (SM2RAIN-ASCAT) is presented. The paper is fairly well written but paints an overly rosy picture of the dataset. Both the dataset and the validation exhibit a number of serious issues which must be addressed before the paper can be published.

(1) The peak underestimation issue has not been resolved in the current release of the dataset, as revealed by both the low STDRATIO values (Figure 3) and the time series comparison (Figure 4). This major issue has been highlighted in two large precipitation dataset evaluations that have been ignored in the present study (https://www.hydrol-earth-syst-sci.net/21/6201/2017/ and https://www.hydrol-earth-syst-sci.net/23/207/2019/). It is important that previously identified issues are addressed or at least discussed.

[Figure]

(2) The CDF correction is based on the REF data and is thus not independent, giving the dataset an unfair advantage compared to GPM-ER in Figure 3.

(3) The RMSE metric should not be used for the evaluation of precipitation datasets at daily time scales as it yields misleading results (makes it seem datasets with underestimated peaks such as SM2RAIN are better). This is due to the high skewness of the precipitation distribution and the prevalence of temporal mismatches between estimated and observed precipitation peaks. The problem is illustrated in the paper by Figure 3, which shows a higher RMSE value (i.e., "worse" performance) for the bias- and CDF-corrected SM2RAIN product (BC-CDF) than for any of the uncorrected SM2RAIN products.

(4) Only correlation and RMSE statistics are presented for the performance evaluation in Figure 6. Please remove the RMSE for the previously mentioned reason and add other metrics, such as variability ratio, bias, hit/miss ratio, frequency of wet days, peak magnitude, etc. for a more thorough performance evaluation.

(5) The TC evaluation only takes into account the monthly correlation – just one aspect of dataset performance (monthly temporal dynamics). Hence the TC evaluation alone cannot be used to conclude whether a particular dataset is better or worse (as is done in the last paragraph of the abstract: "SM2RAIN-ASCAT dataset provides better performance better than GPM and GPCC in the data scarce regions of the world"). Other aspects should also be considered.

(6) "The recent "bottom up" approach that uses satellite soil moisture observations for estimating rainfall through the SM2RAIN algorithm is suited to build long-term and consistent rainfall data record as a single polar orbiting satellite sensor is used." If this is true, why does the dataset span such a short period (2007-2018)? All datasets listed in Table 1 (excluding IMERG) span a longer period. This statement should be revised.

(7) On a related note, the evaluation of https://www.hydrol-earth-syst-sci.net/21/6201/2017/ (co-authored by the first author of the present study) shows that

[Figure]

SM2RAIN-ASCAT performs worst among all precipitation datasets in terms of trend, due to the combination of data from different ASCAT sensors. So are the different ASCAT sensors consistent with each other or not? Has this trend issue been resolved in this SM2RAIN-ASCAT release? If so, this should be shown. If not, this should be communicated to the reader.

(8) In the interest of transparency the abstract should mention that the presented SM2RAIN dataset i) is limited to liquid precipitation (snowfall is not present in the dataset), ii) exhibits spurious drizzle, iii) underestimates extremes (as demonstrated by Figures 3 and 4 of the paper), and iv) potentially suffers from intercalibration issues (see comment (7)). If any of these problems have been fixed in the current release of SM2RAIN-ASCAT, this should be shown in the paper.

(9) "The limitations of the bottom up approach are the possibility to estimate only terrestrial rainfall and its dependence on land characteristics (e.g., low accuracy for dense vegetation coverage and complex topography, Broccaet al., 2014)." The other limitations (spurious drizzle, underestimation of extremes, and intercalibration issues) should also be mentioned here.

(10) To my understanding the regional evaluation is performed using daily accumulations, while the triple collocation (TC) analysis is performed using monthly accumulations – correct? To avoid confusion, please state the time scale of each specific evaluation/analysis in both the abstract and the captions of all figures.

(11) Version numbers should be assigned to the different SM2RAIN-ASCAT releases, to avoid confusion. I know there have been at least two releases. Which one is this?

(12) Please add ERA5 to Figure 3 and make it easier to see the differences among the boxes, either by reducing the range of the y-axes or by exanding the size of the y-axes.

(13) The intro/methods part of the abstract is a bit too long, while the results/discussion part is a bit too short (just three sentences).

(14) "the surface runoff rate, i.e., the water that does not infiltrate into the soil and flows at the surface to the watercourses, is much lower than the rainfall rate, mainly if equation (1) is applied at coarse spatial resolution (20 km), i.e., with satellite soil moisture data." This statement does not make sense to me. Runoff can be equal to rainfall if the soil is saturated, at all scales – from hillslope to catchment.

---

## Referee Comment (RC3) · Anonymous Referee #3 · 6 Jun 2019

This study provides the descriptions and validation results of the 11-year (2007-2018) SM2RAIN-ASCAT global rainfall dataset. Overall, the study fits the scope of ESSD, the paper is well written and the presentation quality is very good. I think that the dataset has great potentials for different applications, as also stated by the authors, especially in specific regions of the world where it seems to outperform other products based on different approaches (e.g., GPM IMERG Early Run). However, there are several aspects that need to be addressed before the paper can be published.

Line 34: What do the author mean by "operationally available in NRT"? This is not a crucial aspect for the dataset presented in this paper.

Line 36: It is important to note that it is not global as it does not provide rainfall over water bodies, and it is limited to the avalability and quality of soil mosture data. This

should be clearly stated also in the conclusions.

Line 42: Please, specify "the IMERG Global Precipitation Measurement (GPM) mission product

Line 75: Rainfall is not "measured" from space. Precipitation retrieval based on "top-down" approaches is very complex due to interaction of the radiation emitted by the Earth's surface with gases, and liquid and solid hydrometeors within the clouds. For example, passive microwave retrieval techniques need to account for variability of all these elements (e.g., surf. emissivity and temperature, water vapour content, cloud water content, sizes, shapes, density, 3-D distribution of liquid, solid and mixed-phase hydrometeors).

Line 76-78: Please, rephrase this sentence: "these methods are based on inversion techniques where the upwelling radiation (or backscattered signal for radars) is related to the surface precipitation rate".

Line 133 (and Line 176, and Line 327): Please, clarify what you mean by "1009 points". Are these 12.5km x1.2km grid boxes? What do you mean by "uniformly distributed? How have they been selected? How many "points" are selected in each region? How are the raingauge measurements treated to be associated to each "point"?

Line 203-205: it is not clear how the 12 hour sampling of the ASCAT soil moisture product is used to obtain the daily (24 hour) SM2RAIN rainfall prouct.

Line 282-284: correction of the overall bias can be very effective for mitigating errors in all products. It should be pointed out by the authors if SM2RAIN-ASCAT dataset presented in this paper is the same product that would be obtained operationally in NRT (see also Line 34). If this is not the case, in my opinion, for a fair comparison, the IMERG GPM Final Run should be used instead of the Early Run in this study. Otherwise, the authors should explain clearly why the GPM Early Run is used in this study. Although I understand that IMERG Final Run can not be used for TC, I recommend to

show the results of SM2RAIN-ASACT compared to IMERG Final Run.

Line 314: Optimal value for FAR is 0, not 1. Please, correct.

Line 316: Please, motivate the choice of 0.5 mm/day (and not a lower value > 0 mm/day) as rainfall/no rainfall threshold.

Line 379-380: How many points are used to compute these averages in each region? Are "problematic" areas for soil moisture retrieval (complex orography, highly vegetated, ecc.) included among the 1009 points used here?

Line 389: Why R and RMSE are considered "more important"? Please, justify this choice.

Line 400-401: It is not clear what periods is used for the calibration in the two separate time frames. I assume that the calibration is not carried out for the whole periods.

Line 404-408: it is not clear what the authors mean by distinguishing "in space" and "in time".

Line 411-413: ERA-5 is used for calibration. It is not fair to use this dataset to create this map, and show R and RMSE.

Line 468: Please, specify what is the committed area for ASCAT products (not ASCAT).

Minor corrections: Line 46: correct: "provides better performance better" Line 100: correct "has the advantage of requiring" Line 138-139: please specify which datasets have been used for the TC, what for the regional assessment, and what for global assessement. Line 190: Please, correct: "spatially averaging" Line 392-393: please correct this sentence. Something is missing, or maybe remove "," after "filtering". Line 402-403: Please, correct this sentence.

---

## Author Comment (AC1) · 11 Jul 2019

**Comments from the Reviewer #1 and the corresponding revision**

**SM2RAIN-ASCAT (2007–2018): GLOBAL DAILY SATELLITE RAINFALL FROM ASCAT SOIL MOISTURE**

by Luca Brocca, Paolo Filippucci, Sebastian Hahn, Luca Ciabatta, Christian Massari, Stefania Camici, Lothar Schüller, Bojan Bojkov, Wolfgang Wagner

**Anonymous Referee #1**

*This study provides detailed descriptions of the SM2Rain product and several evaluation results. Overall, the study would be useful for current and future SM2Rain users, and fits the scope of ESSD. However, I do find that the manuscript misses several key information in SM2Rain production and evaluation.*

We thank the reviewer for her/his appreciation of our study and for the valuable suggestions that helped us to clarify and improve the manuscript. A detailed answer to each comment is reported in the sequel clarifying better the procedure used for developing the global SM2RAIN-ASCAT product.

As general reply to all the reviewers, we would like to underline that the paper goal is to present and describe the SM2RAIN-ASCAT global rainfall dataset and to perform a preliminary assessment of the product with respect to other state-of-the-art global rainfall products. We do not want to show a comprehensive assessment of the product. Indeed, we believe that the validation of the dataset should be performed by researchers other than the product developers (indeed the dataset is made freely available and a first paper was already published: Paredes-Trejo et al., 2019; doi:10.3390/rs11091113). Even better, we stress the importance of performing the validation by using the dataset in hydrological or agricultural applications (e.g., flood prediction and agricultural water management). The comparison with raingauges or any reference dataset could be misleading, mainly when the rainfall products include the ground observed information used for their derivation.

*1. Line 24 – 29: The statement here is too strong. I agree that SM2Rain is a useful product in some aspects. However, I have not seen strong evidences that SM2Rain substantially outperforms other merged products, e.g., MSWEP v2.0. Additionally, soil moisture retrievals prior 2002 have very low data quality. I personally doubt if good precipitation can be derived from these soil moisture data sets. Hence, I also suspect whether SM2Rain "is suited to build long-term consistent rainfall".*

The reviewer is right; the SM2RAIN-ASCAT rainfall product will hardly outperform merged products, mainly if the comparison with raingauges is carried out (see comment above). Apart the possibility to include SM2RAIN-ASCAT in merged products, we believe that a strong added-value of SM2RAIN-ASCAT is its expected availability in the next 25 years, with already 12 years of data available, and its independence with respect to the others state-of-the-art satellite rainfall products (e.g., GPM IMERG, PERSIANN, CMORPH). The sentence was misleading as we intended to say that a long-term SM2RAIN-ASCAT dataset, starting from 2007, and it is ensured until mid-2040s, can be built with the

proposed approach. The sentence will be modified in the revised manuscript, accordingly.

*2. Line 147: A global map of ASCAT temporal sampling frequency would be helpful.*

A global map of ASCAT temporal frequency will be added in the Appendix.

*3. Line 201 – 203: I'm wondering if there are any risks of increasing false rainfall events by linear interpolation?*

The reviewer is right; linear interpolation may increase the risk of false rainfall events, but we didn't find better alternative so far. This comment will be added in the revised manuscript.

*4. Line 242: The authors state that runoff at 20km grid is negligible. Can you provide some rainfall-runoff simulation works to support this hypothesis?*

We are saying that surface runoff is expected to be negligible at larger spatial scales due to the possibility that locally generated surface runoff (e.g., over impervious surfaces) can re-infiltrate into more permeable areas in the same pixel. Of course, this hypothesis can be not valid in some areas, but we have indirectly validated this hypothesis as we have hardly seen the ASCAT soil moisture signal to be completely saturated for more than one day. Therefore, surface runoff due to saturated soil is expected to occur very rarely at 20 km scale. This aspect will be clarified in the revised manuscript showing the number of times ASCAT soil moisture signal is saturated for more than one day.

*5. Line 249: I'm a little bit confused by equation 2. First, the authored stated that ET is negligible. Then, why it is still considered in equation 2? Second, it seems g(t) and e(t) should be plus in sign, according to equation (1)?*

The reviewer is right; g(t) and e(t) must be plus in sign, thanks for spotting the error. In previous applications of SM2RAIN, we have assumed ET negligible, during rainfall, but in this study we wanted to test the possibility to include the ET term and to assess its impact for rainfall estimation through SM2RAIN. For that, we left the e(t) component in equation (2).

*6. Line 261: e(t) is calculated using ERA5 ET. The ERA5 ET is expected to depend on ERA5 precipitation. For instance, a dry period seen by ERA5 (precipitation deficiency) will lead to low ET. Therefore, the authors should discuss the dependency of ERA5 and SM2Rain rainfall product, particularly when TC is considered in the later part of the paper.*

We agree with the reviewer, some dependencies between ERA5 ET and precipitation may occur. However, we underline that in the selected configuration (see lines 385-390) the ERA5 ET is not used and, hence, this dependency is excluded. Moreover, in Triple Collocation Analysis application we didn't consider ERA5, to avoid any dependency between the products (see lines 446-447). These points will be underlined better in the revised manuscript.

*7. Line 291: What is the reference rainfall?*

The reference rainfall is the one used for the calibration of SM2RAIN parameter values and the climatological correction factor. In the section "4.1 Selection of the best SM2RAIN processing configuration at 1009 points", we have used ground-based rainfall observations as reference and it will

be clarified in the revised manuscript. As stated at line 392, for the global SM2RAIN-ASCAT dataset production we have used ERA5 rainfall dataset as reference.

*8. Section 3.4: Please also specify the error model used in this TC analysis.*

As in Massari et al. (2017), we have used an additive error model in TC analysis, it will be clarified in the revised manuscript.

*9. Section 3.5: Equations of these scores will be helpful here.*

The equations will be added in the revised manuscript.

*10. Line 367: I'm still not clear which product is used as a reference to correct SM2Rain.*

In the section "4.1 Selection of the best SM2RAIN processing configuration at 1009 points", the ground-based rainfall observations are used as reference. Differently, in the sections "4.2 Generation of SM2RAIN-ASCAT dataset" and "4.3 Regional and global assessment of SM2RAIN-ASCAT dataset", the ERA5 rainfall is used as reference (see line 392). It will be clarified better in the revised manuscript.

*11. Section 4.1 and Line 385: It's un clear how SM2Rain parameters were calibrated (determined) and extended to the global scale.*

SM2RAIN parameter values are calibrated point-by-point by using the reference rainfall as target (see reply to comment 10 for the definition of reference rainfall). As objective function we have used the minimization of the RMSE between SM2RAIN-ASCAT and reference rainfall datasets. There is no linkage between the local scale and global scale calibration, as different reference rainfall and data periods are used in the two calibrations. It will be clarified better in the revised manuscript.

*12. Figure 5: SM2Rain is calibrated against ERA5. Therefore, the consistency of ERA5 and SM2Rain only suggests how well ASCAT was fitted to ERA5. The authors should be clear that this is not suggesting the accuracy or the performance of SM2Rain (Line 414 – 415).*

The reviewer is right; we will remove the terms "performance" and "accuracy" from this section to avoid misunderstanding. We will underline in the revised manuscript that Figure 5 shows the consistency of ERA5 and SM2RAIN-ASCAT. Of course, we expect better performance in the areas in which the consistency is higher, but the preliminary assessment of SM2RAIN-ASCAT is performed in section 4.3.

*13. Line 431 – 433: SM2Rain show better performances relative to which product? It seems that SM2Rain's R is much lower than the other three in Figure 6 a and b.*

Here, we wanted to highlight the regions where SM2RAIN-ASCAT is performing better, not with respect to other products, but only spatially relative to different regions. It will be clarified in the revised manuscript.

*14. Following the comment above, SM2Rain was derived by calibrations against ERA5. However, its performances are consistently lower than ERA5. Then, what's the contribution/ value of SM2Rain?*

There are several important differences between SM2RAIN-ASCAT and ERA5. The most important difference is the possibility to provide SM2RAIN-ASCAT rainfall in near real time (e.g., with latency

lower than 6 hours), while ERA5 is provided with a latency of weeks. Therefore, SM2RAIN-ASCAT can be used in many applications that require rainfall data with short latency, whereas ERA5 (or GPCC) cannot be used. Moreover, we should underline that ERA5 is using ground observations and in the regions analysed in Figure 6 a dense coverage of ground stations is available. Differently, in poorly gauged areas (e.g., Africa and South America) a lower performance of ERA5 might be expected.

*15. Line 446: What products are used for TC analysis? Massari used ERA. However, I don't think this is appropriate for this study. SM2Rain here is calibrated against ERA, and they may have cross-correlated errors.*

The reviewer is right; we didn't use ERA5 but GPCC, GPM Early Run and SM2RAIN-ASCAT as stated at lines 446-447.

---

## Author Comment (AC2) · 11 Jul 2019

**Comments from the Reviewer #2 and the corresponding revision**

**SM2RAIN-ASCAT (2007–2018): GLOBAL DAILY SATELLITE RAINFALL FROM ASCAT SOIL MOISTURE**

by Luca Brocca, Paolo Filippucci, Sebastian Hahn, Luca Ciabatta, Christian Massari, Stefania Camici, Lothar Schüller, Bojan Bojkov, Wolfgang Wagner

**Anonymous Referee #2**

*An update of a satellite soil moisture-based rainfall dataset (SM2RAIN-ASCAT) is presented. The paper is fairly well written but paints an overly rosy picture of the dataset. Both the dataset and the validation exhibit a number of serious issues which must be addressed before the paper can be published.*

We thank the reviewer for the valuable suggestions that helped us to clarify and improve the manuscript. A detailed answer to each comment is reported in the sequel.

As general reply to all the reviewers, we would like to underline that the paper goal is to present and describe the SM2RAIN-ASCAT global rainfall dataset and to perform a preliminary assessment of the product with respect to other state-of-the-art global rainfall products. We do not want to show a comprehensive assessment of the product. Indeed, we believe that the validation of the dataset should be performed by researchers other than the product developers (indeed the dataset is made freely available and a first paper was already published: Paredes-Trejo et al., 2019; doi:10.3390/rs11091113). Even better, we stress the importance of performing the validation by using the dataset in hydrological or agricultural applications (e.g., flood prediction and agricultural water management). The comparison with raingauges or any reference dataset could be misleading, mainly when the rainfall products include the ground observed information used for their derivation.

*(1) The peak underestimation issue has not been resolved in the current release of the dataset, as revealed by both the low STDRATIO values (Figure 3) and the time series comparison (Figure 4). This major issue has been highlighted in two large precipitation dataset evaluations that have been ignored in the present study (https://www.hydrol-earth-syst-sci.net/21/6201/2017/ and https://www.hydrolearth-syst-sci.net/23/207/2019/). It is important that previously identified issues are addressed or at least discussed.*

The reviewer is right; SM2RAIN-ASCAT has underestimation issue that has not been resolved completely. It will be clearly underlined in the revised manuscript. However, we want to stress that the climatological correction partly addresses this issue. A more specific CDF correction can be used for addressing the target (e.g., daily CFD matching), but we have preferred not to implement to avoid a strong dependency between SM2RAIN-ASCAT rainfall data and the reference dataset (indeed, also the reference might be wrong, particularly in poorly gauged regions).

*(2) The CDF correction is based on the REF data and is thus not independent, giving the dataset an unfair advantage compared to GPM-ER in Figure 3.*

In figure 3, we have used GMP-ER as a state-of-the-art reference, not to perform a comparison between the datasets. As mentioned above, the paper is not intended to perform a comprehensive assessment of SM2RAIN-ASCAT dataset, or its comparison in terms of accuracy with respect to other products. We only want to show that SM2RAIN-ASCAT is performing similarly to state-of-the-art products and, hence, can be a valuable alternative for applications using rainfall observations as input. It will be specified better in the revised manuscript.

*(3) The RMSE metric should not be used for the evaluation of precipitation datasets at daily time scales as it yields misleading results (makes it seem datasets with underestimated peaks such as SM2RAIN are better). This is due to the high skewness of the precipitation distribution and the prevalence of temporal mismatches between estimated and observed precipitation peaks. The problem is illustrated in the paper by Figure 3, which shows a higher RMSE value (i.e., "worse" performance) for the bias- and CDF-corrected SM2RAIN product (BC-CDF) than for any of the uncorrected SM2RAIN products.*

We agree with the reviewer that the RMSE statistic has some limitations in evaluating precipitation datasets. Indeed, we have used different statistics and in the revised manuscript we will perform the evaluation with multiple statistics also for Figure 6. Anyhow, RMSE is used in many papers evaluating precipitation datasets (and we don't believe they are all wrong), and suffers from similar limitation as any single score used for assessing a dataset. As mentioned above, we believe that the real validation should be performed using the rainfall products in the hydrological or agricultural applications. These aspects will be underlined in the revised manuscript.

*(4) Only correlation and RMSE statistics are presented for the performance evaluation in Figure 6. Please remove the RMSE for the previously mentioned reason and add other metrics, such as variability ratio, bias, hit/miss ratio, frequency of wet days, peak magnitude, etc. for a more thorough performance evaluation.*

In the revised manuscript we will add multiple statistics in Figure 6, similarly to Figure 3.

*(5) The TC evaluation only takes into account the monthly correlation – just one aspect of dataset performance (monthly temporal dynamics). Hence the TC evaluation alone cannot be used to conclude whether a particular dataset is better or worse (as is done in the last paragraph of the abstract: "SM2RAIN-ASCAT dataset provides better performance better than GPM and GPCC in the data scarce regions of the world"). Other aspects should also be considered.*

TC analysis is performed at daily time scale, not monthly time scale. Therefore, we believe TC analysis provides some information on the accuracy of the different rainfall products at daily time scale, it will be clarified in the revised manuscript.

*(6) "The recent "bottom up" approach that uses satellite soil moisture observations for estimating rainfall through the SM2RAIN algorithm is suited to build long-term and consistent rainfall data record as a single polar orbiting satellite sensor is used." If this is true, why does the dataset span such a short*

*period (2007-2018)? All datasets listed in Table 1 (excluding IMERG) span a longer period. This statement should be revised.*

The statement will be revised as we intended to say that a long-term SM2RAIN-ASCAT dataset, starting from 2007, and it is ensured until mid-2040s, can be built based the proposed approach. Sorry for the misunderstanding that will be corrected in the revised manuscript.

*(7) On a related note, the evaluation of https://www.hydrol-earth-systsci.net/21/6201/2017/ (co-authored by the first author of the present study) shows that SM2RAIN-ASCAT performs worst among all precipitation datasets in terms of trend, due to the combination of data from different ASCAT sensors. So are the different ASCAT sensors consistent with each other or not? Has this trend issue been resolved in this SM2RAIN-ASCAT release? If so, this should be shown. If not, this should be communicated to the reader.*

The trend issue has been solved as in the previous delivery of the SM2RAIN-ASCAT dataset (preliminary distribution) we didn't consider appropriately the availability of two ASCAT sensors (Metop-A and -B) after 2013. The dual calibration performed in this study (see lines 400-402) has been carried out exactly to address this issue. It will be clarified in the revised manuscript.

*(8) In the interest of transparency the abstract should mention that the presented SM2RAIN dataset i) is limited to liquid precipitation (snowfall is not present in the dataset), ii) exhibits spurious drizzle, iii) underestimates extremes (as demonstrated by Figures 3 and 4 of the paper), and iv) potentially suffers from intercalibration issues (see comment (7)). If any of these problems have been fixed in the current release of SM2RAIN-ASCAT, this should be shown in the paper.*

As suggested by the reviewer, we will clearly communicate the limitations of SM2RAIN-ASCAT dataset in the abstract of the revised manuscript. Limitations and strengths of the SM2RAIN-ASCAT dataset will be clearly illustrated. In the interest of transparency, we have made the SM2RAIN-ASCAT product freely available, and also the dataset at 1009 points that we have used for selecting the best configuration to develop the product. All the input and test datasets used in the paper are freely available and the analysis can be easily performed by the reader (note that also SM2RAIN code is made available on GitHub).

*(9) "The limitations of the bottom up approach are the possibility to estimate only terrestrial rainfall and its dependence on land characteristics (e.g., low accuracy for dense vegetation coverage and complex topography, Broccaet al., 2014)." The other limitations (spurious drizzle, underestimation of extremes, and intercalibration issues) should also be mentioned here.*

Limitations and strengths of the SM2RAIN-ASCAT dataset will be clearly illustrated in the revised manuscript.

*(10) To my understanding the regional evaluation is performed using daily accumulations, while the triple collocation (TC) analysis is performed using monthly accumulations – correct? To avoid*

*confusion, please state the time scale of each specific evaluation/analysis in both the abstract and the captions of all figures.*

All the analyses have been performed at daily time scale and it will be clarified in the revised manuscript.

*(11) Version numbers should be assigned to the different SM2RAIN-ASCAT releases, to avoid confusion. I know there have been at least two releases. Which one is this?*

The first official version of the SM2RAIN-ASCAT dataset can be considered the one presented in this paper. Indeed, the dataset has been published on Zenodo and a DOI (digital object identifier) has been assigned to the dataset to avoid confusion.

*(12) Please add ERA5 to Figure 3 and make it easier to see the differences among the boxes, either by reducing the range of the y-axes or by exanding the size of the y-axes.*

In the revised manuscript we will add ERA5 and we will also improve figure readability.

*(13) The intro/methods part of the abstract is a bit too long, while the results/discussion part is a bit too short (just three sentences).*

The abstract of the revised manuscript will be revised accordingly.

*(14) "the surface runoff rate, i.e., the water that does not infiltrate into the soil and flows at the surface to the watercourses, is much lower than the rainfall rate, mainly if equation (1) is applied at coarse spatial resolution (20 km), i.e., with satellite soil moisture data." This statement does not make sense to me. Runoff can be equal to rainfall if the soil is saturated, at all scales – from hillslope to catchment.*

We are saying that surface runoff is expected to be negligible at larger spatial scales due to the possibility that locally generated surface runoff (e.g., over impervious surfaces) can re-infiltrate into more permeable areas in the same pixel. Of course, this hypothesis can be not valid in some areas, but we have indirectly validated this hypothesis as we have hardly seen the ASCAT soil moisture signal to be completely saturated for more than one day. Therefore, surface runoff due to saturated soil is expected to occur very rarely at 20 km scale. This aspect will be clarified in the revised manuscript showing the number of times ASCAT soil moisture signal is saturated for more than one day.

---

## Author Comment (AC3) · 11 Jul 2019

**Comments from the Reviewer #3 and the corresponding revision**

**SM2RAIN-ASCAT (2007–2018): GLOBAL DAILY SATELLITE RAINFALL FROM ASCAT SOIL MOISTURE**

by Luca Brocca, Paolo Filippucci, Sebastian Hahn, Luca Ciabatta, Christian Massari, Stefania Camici, Lothar Schüller, Bojan Bojkov, Wolfgang Wagner

**Anonymous Referee #3**

*This study provides the descriptions and validation results of the 11-year (2007-2018) SM2RAIN-ASCAT global rainfall dataset. Overall, the study fits the scope of ESSD, the paper is well written and the presentation quality is very good. I think that the dataset has great potentials for different applications, as also stated by the authors, especially in specific regions of the world where it seems to outperform other products based on different approaches (e.g., GPM IMERG Early Run). However, there are several aspects that need to be addressed before the paper can be published.*

We thank the reviewer for her/his appreciation of our study and for the valuable suggestions that helped us to clarify and improve the manuscript. A detailed answer to each comment is reported in the sequel.

As general reply to all the reviewers, we would like to underline that the paper goal is to present and describe the SM2RAIN-ASCAT global rainfall dataset and to perform a preliminary assessment of the product with respect to other state-of-the-art global rainfall products. We do not want to show a comprehensive assessment of the product. Indeed, we believe that the validation of the dataset should be performed by researchers other than the product developers (indeed the dataset is made freely available and a first paper was already published: Paredes-Trejo et al., 2019; doi:10.3390/rs11091113). Even better, we stress the importance of performing the validation by using the dataset in hydrological or agricultural applications (e.g., flood prediction and agricultural water management). The comparison with raingauges or any reference dataset could be misleading, mainly when the rainfall products include the ground observed information used for their derivation.

*Line 34: What do the author mean by "operationally available in NRT"? This is not a crucial aspect for the dataset presented in this paper.*

The reviewer is right; the sentence will be removed by the abstract in the revised manuscript. It is due to a parallel activity we are performing for producing a NRT SM2RAIN-ASCAT product.

*Line 36: It is important to note that it is not global as it does not provide rainfall over water bodies, and it is limited to the avalability and quality of soil mosture data. This should be clearly stated also in the conclusions.*

The reviewer is right; SM2RAIN-ASCAT product is not global and it will be clarified in the abstract and in the conclusions of the revised manuscript.

*Line 42: Please, specify "the IMERG Global Precipitation Measurement (GPM) mission product*

The text will be modified, accordingly.

*Line 75: Rainfall is not "measured" from space. Precipitation retrieval based on "topdown" approaches is very complex due to interaction of the radiation emitted by the Earth's surface with gases, and liquid and solid hydrometeors within the clouds. For example, passive microwave retrieval techniques need to account for variability of all these elements (e.g., surf. emissivity and temperature, water vapour content, cloud water content, sizes, shapes, density, 3-D distribution of liquid, solid and mixed-phase hydrometeors).*

The term "measured" will be removed from the revised manuscript, even though it's matter of terminology. Every measurement is affected by errors.

*Line 76-78: Please, rephrase this sentence: "these methods are based on inversion techniques where the upwelling radiation (or backscattered signal for radars) is related to the surface precipitation rate".*

The sentence will be revised.

*Line 133 (and Line 176, and Line 327): Please, clarify what you mean by "1009 points". Are these 12.5km x1.2km grid boxes? What do you mean by "uniformly distributed? How have they been selected? How many "points" are selected in each region? How are the raingauge measurements treated to be associated to each "point"?*

The 1009 points are uniformly distributed over a regular grid with spacing of 1.5°. Each point is considered representative of a 0.25° x 0.25° box; the selection is carried out for reducing the computational time for running the different SM2RAIN configurations. The numbers of points for each region is based on the size of the region (328 points in Australia, 163 in India, 55 in Italy, and 463 in the United States). Ground observations and GPM-ER data are regridded by spatial averaging measurements contained over each 0.25° x 0.25° box. All these details will be reported in the revised manuscript.

*Line 203-205: it is not clear how the 12 hour sampling of the ASCAT soil moisture product is used to obtain the daily (24 hour) SM2RAIN rainfall prouct.*

The 24-hour accumulated rainfall is obtained by summing the two 12-hour accumulated rainfall data obtained for each day, it will be specified in the revised manuscript.

*Line 282-284: correction of the overall bias can be very effective for mitigating errors in all products. It should be pointed out by the authors if SM2RAIN-ASCAT dataset presented in this paper is the same product that would be obtained operationally in NRT (see also Line 34). If this is not the case, in my opinion, for a fair comparison, the IMERG GPM Final Run should be used instead of the Early Run in this study. Otherwise, the authors should explain clearly why the GPM Early Run is used in this study. Although I understand that IMERG Final Run can not be used for TC, I recommend to show the results of SM2RAIN-ASCAT compared to IMERG Final Run.*

The SM2RAIN-ASCAT dataset presented in the paper is the same product that would be obtained operationally in NRT. The climatological correction is performed with constant parameter values and,

hence, it can be implemented in NRT. We note that a climatological correction is performed in several satellite rainfall datasets delivered in NRT (e.g., 3B42RT, IMERG ER, PERSIANN CCS, CMORPH CRT).

*Line 314: Optimal value for FAR is 0, not 1. Please, correct.*

The reviewer is right; we will correct the error in the revised manuscript, thanks for spotting the mistake.

*Line 316: Please, motivate the choice of 0.5 mm/day (and not a lower value > 0 mm/day) as rainfall/no rainfall threshold.*

As mentioned in the manuscript, the threshold is selected in order to exclude spurious events that might be due to rainfall interpolation\regridding in the reference datasets.

*Line 379-380: How many points are used to compute these averages in each region? Are "problematic" areas for soil moisture retrieval (complex orography, highly vegetated, ecc.) included among the 1009 points used here?*

All points in each region are used, i.e., 328 points in Australia, 163 in India, 55 in Italy, and 463 in the United States. The "problematic" areas are included as 1009 points are randomly selected; no masking has been carried out in this analysis.

*Line 389: Why R and RMSE are considered "more important"? Please, justify this choice.*

We believe that R and RMSE are the two most important statistics for evaluating precipitation datasets after performing several assessment studies of different datasets. However, we acknowledge that the selection of the statistics could be arbitrary and in the revised manuscript we will add multiple statistics at Figure 6 (similarly to Figure 3) to provide a more comprehensive assessment of the products.

*Line 400-401: It is not clear what periods is used for the calibration in the two separate time frames. I assume that the calibration is not carried out for the whole periods.*

In the development of the global SM2RAIN-ASCAT dataset the calibration is performed for the whole periods. Indeed, we do not want to perform calibration and validation against ERA5. As mentioned above, the validation should be performed with independent datasets, and even better by using the product for applications.

*Line 404-408: it is not clear what the authors mean by distinguishing "in space" and "in time".*

In space, we mean a fixed spatial mask over which we are aware of the lower performance of the ASCAT soil moisture product, and consequently of SM2RAIN-ASCAT rainfall product. In time, we have considered a temporally variable mask that flags observations with soil temperature, obtained from ERA5, lower than 3°C. It will be specified better in the revised manuscript.

*Line 411-413: ERA-5 is used for calibration. It is not fair to use this dataset to create this map, and show R and RMSE.*

The reviewer is right; Figure 5 shows the consistency of ERA5 and SM2RAIN-ASCAT and not the

"accuracy" or the "performance" of the product, these terms will be removed from this section of the revised manuscript. Of course, we expect better performance in the areas in which the consistency is higher, but the preliminary assessment of SM2RAIN-ASCAT is performed in section 4.3.

*Line 468: Please, specify what is the committed area for ASCAT products (not ASCAT).*

The reviewer is right; the committed area refers to the ASCAT soil moisture product; it will be specified in the revised manuscript.

***Minor corrections:***

*Line 46: correct: "provides better performance better"*

The text will be modified, accordingly.

*Line 100: correct "has the advantage of requiring"*

The text will be modified, accordingly.

*Line 138-139: please specify which datasets have been used for the TC, what for the regional assessment, and what for global assessement.*

The datasets used for the three analyses will be specified in the revised manuscript.

*Line 190: Please, correct: "spatially averaging"*

The text will be modified, accordingly.

*Line 392-393: please correct this sentence. Something is missing, or maybe remove "," after "filtering".*

The sentence will be revised, accordingly.

*Line 402-403: Please, correct this sentence.*

The sentence will be corrected.

---

## Author Response (AR1)

**Comments from the Reviewers and the corresponding revision**

**SM2RAIN-ASCAT (2007–2018): GLOBAL DAILY SATELLITE RAINFALL FROM ASCAT SOIL MOISTURE**

by Luca Brocca, Paolo Filippucci, Sebastian Hahn, Luca Ciabatta, Christian Massari, Stefania Camici, Lothar Schüller, Bojan Bojkov, Wolfgang Wagner

We thank the reviewers for their appreciation of our study and for the valuable suggestions that helped us to clarify and improve the manuscript. A detailed answer to each comment is reported in the sequel clarifying the procedure used for developing the global SM2RAIN-ASCAT data record and the obtained results. In Italic, we have reported the reviewers comments, in blue, the detailed replies, and in red, the text changed and/or added in the revised manuscript to address reviewers' comments.

As general reply to all the reviewers, we would like to underline that the paper goal is to present and describe the SM2RAIN-ASCAT global rainfall data record and to perform a comparison with state-of-the-art global rainfall products. We do not want to show a comprehensive assessment of the product. Indeed, we believe that researchers other than the product developers should perform the assessment and the validation of the dataset. This clarification has been added in the revised manuscript at the end of the Introduction section (see lines 154-160):

"We underline that the paper goal is to present and describe the SM2RAIN-ASCAT quasi-global rainfall data record and to perform a comparison with state-of-the-art global rainfall products. We do not want to show a comprehensive assessment of the product. Indeed, we believe that researchers other than the product developers should perform the validation of the dataset. Even better, we stress the importance of performing the validation by using the datasets in hydrological or agricultural applications (e.g., flood prediction and agricultural water management)."

Indeed, the dataset is made freely available and a first paper has been already published by Paredes-Trejo et al., 2019 (doi:10.3390/rs11091113) who have assessed the accuracy of the SM2RAIN-ASCAT data record in Brazil. Even better, we stress the importance of performing the validation by using the data record in hydrological or agricultural applications. The comparison with raingauges or any reference dataset could be misleading, mainly when the rainfall products include the ground observed information used for their derivation. Another independent paper has been just submitted by Mazzoleni et al. (2019) who have performed the hydrological validation of SM2RAIN-ASCAT in 8 large basins worldwide showing that the product outperforms all the other satellite-only rainfall dataset. The paper preprint is available on EarthArXiv at https://eartharxiv.org/v2r7c/.

**Anonymous Referee #1**

*This study provides detailed descriptions of the SM2Rain product and several evaluation results. Overall, the study would be useful for current and future SM2Rain users, and fits the scope of ESSD. However, I do find that the manuscript misses several key information in SM2Rain production and evaluation.*

We thank the reviewer for her/his appreciation of our study and for the valuable suggestions that helped us to clarify and improve the manuscript. A detailed answer to each comment is reported in the sequel clarifying better the procedure used for developing the global SM2RAIN-ASCAT product.

*1. Line 24 – 29: The statement here is too strong. I agree that SM2Rain is a useful product in some aspects. However, I have not seen strong evidences that SM2Rain substantially outperforms other merged products, e.g., MSWEP v2.0. Additionally, soil moisture retrievals prior 2002 have very low data quality. I personally doubt if good precipitation can be derived from these soil moisture data sets. Hence, I also suspect whether SM2Rain "is suited to build long-term consistent rainfall".*

The reviewer is right; the SM2RAIN-ASCAT rainfall product will hardly outperform merged products, mainly if the comparison with raingauges is carried out (see the general answer above). Apart the possibility to include SM2RAIN-ASCAT in merged products, we believe that a strong added-value of SM2RAIN-ASCAT is its expected availability in the next 25 years, with already 12 years of data available, and its independence with respect to the others state-of-the-art satellite rainfall products (e.g., GPM IMERG, PERSIANN, CMORPH). The sentence was misleading as we intended to say that a long-term SM2RAIN-ASCAT dataset, starting from 2007, and ensured until mid-2040s, can be built with the proposed approach. The sentence has been modified in the revised manuscript, accordingly (see lines 29-33):

"We exploit here the Advanced SCATterometer (ASCAT) on board three Metop satellites, launched in 2006, 2012 and 2018, as part of the EUMETSAT Polar System programme. The continuity of the scatterometer sensor is ensured until mid-2040s through the Metop Second Generation Programme. By applying SM2RAIN algorithm to ASCAT soil moisture observations, a long-term rainfall data record will be obtained, starting in 2007 until mid-2040s."

*2. Line 147: A global map of ASCAT temporal sampling frequency would be helpful.*

A global map of ASCAT temporal sampling frequency has been added in the Appendix (see figure A1 and lines 218-220):

"(see ***Figure A1*** for the mean daily revisit time of ASCAT in the period 2007-2012 with only Metop-A and the period 2013-2018 with Metop-A+B)"

[Figure]

**Figure A1.** Mean daily revisit time [days] of ASCAT soil moisture observations for the period 2007-2012 (only Metop-A, top panel) and for the period 2013-2018 (Metop-A+B, bottom panel).

*3. Line 201 – 203: I'm wondering if there are any risks of increasing false rainfall events by linear interpolation?*

The reviewer is right; linear interpolation may increase the risk of false rainfall events, and future research will be addressed to mitigate this problem. The comment has been added in the revised manuscript at lines 222-223:

"The interpolation may increase the risk of false rainfall events, but it is a required step to obtain accumulated rainfall over a fixed duration."

*4. Line 242: The authors state that runoff at 20km grid is negligible. Can you provide some rainfall-runoff simulation works to support this hypothesis?*

With the runoff assumption, we are saying that surface runoff is expected to be negligible at larger spatial scales due to the possibility that locally generated surface runoff (e.g., over impervious surfaces) can re-infiltrate into more permeable areas in the same pixel. Of course, this hypothesis can be not valid in some areas, but we have indirectly validated this hypothesis as we have hardly seen the ASCAT soil moisture signal to be saturated for more than one day. Therefore, surface runoff due to saturated soil is expected to occur very rarely at 20 km scale. This aspect has been clarified in the revised manuscript showing the number of days the ASCAT soil moisture signal is saturated for more than one day (see lines 269-277):

"We have indirectly tested this hypothesis by counting the number of days the ASCAT soil moisture product is higher than 99.5 percentile for two (or more) consecutive days in the period 2007-2018. We have found that the number of consecutive days in which the soil is saturated is equal to 4 days (median value on a global scale) over 12 years, with 90% of land pixels with values lower than 12 days (i.e., 1 day per year). The occurrence of higher values is limited to very few areas in the tropical forests and over Himalaya (see Figure A2)."

[Figure]

**Figure A2.** Number of days in which ASCAT soil moisture observations are close to saturation (>99.5 percentile, top panel) for 2 (or more) consecutive days in the period 2007-2018. Please note that the upper value is set to 20 days as in most of land areas the occurrence is very low (90% of land pixel with values lower than 12 days over 12 years). In the bottom panel the soil moisture values at 99.5 percentile (in the period 2007-2018) are shown.

*5. Line 249: I'm a little bit confused by equation 2. First, the authored stated that ET is negligible. Then, why it is still considered in equation 2? Second, it seems g(t) and e(t) should be plus in sign, according to equation (1)?*

The reviewer is right; g(t) and e(t) must be plus in sign, thanks for spotting the error that we have corrected in the revised manuscript (see line 281). In previous applications of SM2RAIN, we have assumed ET negligible, during rainfall, but in this study we wanted to test the possibility to include the

ET term and to assess its impact for rainfall estimation through SM2RAIN. For that, we left the e(t) component in equation (2) but we have used this formulation only at the analysis over 1009 points.

*6. Line 261: e(t) is calculated using ERA5 ET. The ERA5 ET is expected to depend on ERA5 precipitation. For instance, a dry period seen by ERA5 (precipitation deficiency) will lead to low ET. Therefore, the authors should discuss the dependency of ERA5 and SM2Rain rainfall product, particularly when TC is considered in the later part of the paper.*

We agree with the reviewer, some dependencies between ERA5 ET and precipitation may occur. However, we underline that in the selected configuration (see lines 434-436 in the revised manuscript) the ERA5 ET is not used and, hence, this dependency is excluded. Moreover, in Triple Collocation Analysis application we didn't consider ERA5, to avoid any dependency between the products. This point has been underlined better in the revised manuscript (see lines 509-510):

"In TC analysis we have not considered ERA5 purposely to avoid any dependency between the products."

*7. Line 291: What is the reference rainfall?*

The reference rainfall is the one used for the calibration of SM2RAIN parameter values and the climatological correction factor. In the section "4.1 Selection of the best SM2RAIN processing configuration at 1009 points", we have used ground-based rainfall observations as reference and it has been clarified in the revised manuscript (see lines 375-377):

"The ground-based high quality rainfall observations available for the four regions are used as reference data (ground truth) in this analysis."

As stated at lines 440-443, for the global SM2RAIN-ASCAT dataset production we have used ERA5 rainfall dataset as reference.

"As reference dataset for calibrating the filtering, SM2RAIN, and post-processing parameter values, the ERA5 rainfall has been used mainly because of its higher spatial resolution compared to GPCC (36 km versus 100 km)."

And in the Introduction at lines 146-153:

"As reference datasets we have used high-quality local raingauge networks from 2013 to 2017 in the United States, Italy, India and Australia for the assessment at 1009 points and for the regional assessment. Three additional global datasets have been considered: the latest reanalysis of the European Centre for Medium-Range Weather Forecasts (ECMWF), ERA5, the gauge-based Global Precipitation Climatology Centre (GPCC), and the GPM IMERG product (Early Run version). ERA5 has been used for the generation of the quasi-global SM2RAIN-ASCAT data record; GPCC and GPM IMERG have been considered for the TC analysis."

*8. Section 3.4: Please also specify the error model used in this TC analysis.*

As in Massari et al. (2017), we have used an additive error model in TC analysis, it has been clarified in the revised manuscript at lines 337-340:

"In this study, we have implemented the same procedure as described in Massari et al. (2017), i.e., by implementing an additive error model at daily time scale, and we refer the reader to this study for the analytical details."

*9. Section 3.5: Equations of these scores will be helpful here.*

The equations have been added in the Table A1 of the Appendix of the revised manuscript.

"For a complete description of the performance scores, see Table A1 in the Appendix."

**Table 1.** Equations used for the performance scores. For the continuous scores, $P_{ref}$ is the reference dataset (e.g., ground observations, ERA5) and $P_{est}$ is the estimated dataset (e.g., SM2RAIN-ASCAT, GPM-ER), *cov* is the covariance operator, $\sigma$ is the standard deviation operator, $\sum$ is the summation operator, and $N$ is the sample size. For the categorical scores, $H$ is the number of successfully predicted events by a given rainfall product, $F$ the number of non-events erroneously predicted to occur, and $M$ the number of actual events that are missed.

| Performance Score | Score symbol | Equation |
|---|---|---|
| **Continuous scores** | | |
| Pearson's correlation | R | $R = \dfrac{cov(P_{est}, P_{ref})}{\sigma(P_{est})\sigma(P_{ref})}$ |
| Root Mean Square Error | RMSE | $RMSE = \sqrt{\dfrac{\sum(P_{est} - P_{ref})^2}{N}}$ |
| Temporal Variability Ratio | STDRATIO | $STDRATIO = \dfrac{\sigma(P_{est})}{\sigma(P_{ref})}$ |
| Bias | BIAS | $BIAS = \dfrac{\sum(P_{est} - P_{ref})}{N}$ |
| **Categorical scores** | | |
| False Alarm Ratio | FAR | $FAR = \dfrac{F}{H + F}$ |
| Probability of Detection | POD | $POD = \dfrac{H}{H + M}$ |

| Threat Score | TS | $TS = \dfrac{H}{H + F + M}$ |
| --- | --- | --- |

*10. Line 367: I'm still not clear which product is used as a reference to correct SM2Rain.*

In the section "4.1 Selection of the best SM2RAIN processing configuration at 1009 points", the ground-based rainfall observations are used as reference. Differently, in the sections "4.2 Generation of SM2RAIN-ASCAT dataset" and "4.3 Regional and global assessment of SM2RAIN-ASCAT dataset", the ERA5 rainfall is used as reference. It has been clarified better in the revised manuscript (see reply to comment (7)).

*11. Section 4.1 and Line 385: It's un clear how SM2Rain parameters were calibrated (determined) and extended to the global scale.*

SM2RAIN parameter values are calibrated point-by-point by using the reference rainfall as target (see reply to comment 10 for the definition of reference rainfall). As objective function we have used the minimization of the RMSE between SM2RAIN-ASCAT and reference rainfall datasets. There is no linkage between the local scale and global scale calibration, as different reference rainfall and data periods are used in the two calibrations. It has been clarified in the revised manuscript (see lines 312-314):

"SM2RAIN parameter values are calibrated point-by-point by using the reference rainfall as target. As objective function, we have used the minimization of the RMSE between SM2RAIN-ASCAT and reference rainfall."

*12. Figure 5: SM2Rain is calibrated against ERA5. Therefore, the consistency of ERA5 and SM2Rain only suggests how well ASCAT was fitted to ERA5. The authors should be clear that this is not suggesting the accuracy or the performance of SM2Rain (Line 414 – 415).*

The reviewer is right; we have removed the terms "performance" and "accuracy" from this section to avoid misunderstanding (see lines 461-478). Of course, we expect better performance in the areas in which the consistency is higher, but the preliminary assessment of SM2RAIN-ASCAT is performed in section 4.3. We have underlined in the revised manuscript that Figure 5 shows the consistency of ERA5 and SM2RAIN-ASCAT (see lines 462-466):

"Therefore, *Figure 5* illustrates the consistency between SM2RAIN-ASCAT and ERA5, and it is not intended to assess the performance of the data record (even though we expect better accuracy in areas where the agreement is higher)."

*13. Line 431 – 433: SM2Rain show better performances relative to which product? It seems that SM2Rain's R is much lower than the other three in Figure 6 a and b.*

Here, we wanted to highlight the regions where SM2RAIN-ASCAT is performing better, not with respect to other products, but only across to different regions (note that we have added different performance metrics in the Figure 6 of the revised manuscript). It has been clarified in the revised manuscript (see lines 484-488):

"By focusing on the SM2RAIN-ASCAT data record performance over the different regions, it shows better performance in Italy (median R=0.67) and United States (median R=0.62), almost comparable with the other datasets; while in Australia and India R-values are lower (median R=0.61 and 0.59)."

*14. Following the comment above, SM2Rain was derived by calibrations against ERA5. However, its performances are consistently lower than ERA5. Then, what's the contribution/ value of SM2Rain?*

There are several important differences between SM2RAIN-ASCAT and ERA5. The most important difference is the possibility to provide SM2RAIN-ASCAT rainfall in near real time (e.g., with latency lower than 6 hours), while ERA5 is provided with a latency of weeks. Therefore, SM2RAIN-ASCAT can be used in many applications that require rainfall data with short latency, whereas ERA5 (or GPCC) cannot be used. It has been underlined at lines 492-494:

"We highlight also that differently from SM2RAIN-ASCAT and GPM-ER, GPCC and ERA5 have a latency of weeks to months and, hence, these products cannot be used for near real time applications."

Moreover, we should underline that ERA5 is using ground observations and in the regions analysed in Figure 6 a dense coverage of ground stations is available. Differently, in poorly gauged areas (e.g., Africa and South America) a lower performance of ERA5 might be expected.

*15. Line 446: What products are used for TC analysis? Massari used ERA. However, I don't think this is appropriate for this study. SM2Rain here is calibrated against ERA, and they may have cross-correlated errors.*

The reviewer is right; we didn't use ERA5 but GPCC, GPM Early Run and SM2RAIN-ASCAT as stated at lines 507-510:

"On a global scale, the TC approach has been implemented by using the triplet SM2RAIN-ASCAT, GPM-ER and GPCC, by considering the common period 2015-2018. In TC analysis we have not considered ERA5 purposely to avoid any dependency between the products."

**Anonymous Referee #2**

*An update of a satellite soil moisture-based rainfall dataset (SM2RAIN-ASCAT) is presented. The paper is fairly well written but paints an overly rosy picture of the dataset. Both the dataset and the validation exhibit a number of serious issues which must be addressed before the paper can be published.*

We thank the reviewer for the valuable suggestions that helped us to clarify and improve the manuscript. A detailed answer to each comment is reported in the sequel.

*(1) The peak underestimation issue has not been resolved in the current release of the dataset, as revealed by both the low STDRATIO values (Figure 3) and the time series comparison (Figure 4). This major issue has been highlighted in two large precipitation dataset evaluations that have been ignored in the present study (https://www.hydrol-earth-syst-sci.net/21/6201/2017/ and https://www.hydrolearth-syst-sci.net/23/207/2019/). It is important that previously identified issues are addressed or at least discussed.*

The reviewer is right; SM2RAIN-ASCAT has underestimation issue that has not been resolved completely. It has been clearly underlined in the revised manuscript (see below). However, we want to stress that the climatological correction partly addresses this issue. A more specific CDF correction can be used for addressing the target (e.g., daily CFD matching), but we have preferred not to implement to avoid a strong dependency between SM2RAIN-ASCAT rainfall data and the reference dataset (indeed, also the reference might be wrong, particularly in poorly gauged regions).

Lines 48-51:

"Limitations of SM2RAIN-ASCAT data record consist in the underestimation of peak rainfall events, in the occurrence of spurious rainfall events due to high frequency soil moisture fluctuations that might be corrected with more advanced bias correction techniques."

Lines 387-390:

"Very good statistics have been obtained in terms of RMSE and BIAS but a tendency to underestimate the observed rainfall variability (median STDRATIO=0.60) and medium-high probability of false alarm (median FAR=0.53). The other scores are similar, or slightly lower than those obtained through GPM-ER."

Lines 501-506:

"As shown also in **Figure 3**, the SM2RAIN-ASCAT data record has limitations in reproducing the variability of rainfall (low STDRATIO) mainly due underestimation issues. Moreover, FAR values of SM2RAIN-ASCAT are quite high highlighting the difficulties in removing the problem of high frequency soil moisture fluctuations wrongly interpreted by SM2RAIN as rainfall events."

Lines 552-555:

"Limitations of SM2RAIN-ASCAT data record consist in: 1) the underestimation of peak rainfall events, 2) the presence of spurious rainfall events due to high frequency soil moisture fluctuations, 3) the estimation of liquid rainfall only (snowfall cannot be estimated), and 4) the possibility to estimate rainfall over land only."

*(2) The CDF correction is based on the REF data and is thus not independent, giving the dataset an unfair advantage compared to GPM-ER in Figure 3.*

In figure 3, we have used GMP-ER as a state-of-the-art reference, not to perform a comparison between the datasets. As mentioned above, the paper is not intended to perform a comprehensive assessment of SM2RAIN-ASCAT dataset, or its comparison in terms of accuracy with respect to other products. We only want to show that SM2RAIN-ASCAT is performing similarly to state-of-the-art products and, hence, can be a valuable alternative for applications using rainfall observations as input. It has been specified better in the revised manuscript at lines 154-160:

"We underline that the paper goal is to present and describe the SM2RAIN-ASCAT quasi-global rainfall data record and to perform a comparison with state-of-the-art global rainfall products. We do not want to show a comprehensive assessment of the product. Indeed, we believe that researchers other than the product developers should perform the validation of the dataset. Even better, we stress the importance of performing the validation by using the datasets in hydrological or agricultural applications (e.g., flood prediction and agricultural water management)."

*(3) The RMSE metric should not be used for the evaluation of precipitation datasets at daily time scales as it yields misleading results (makes it seem datasets with underestimated peaks such as SM2RAIN are better). This is due to the high skewness of the precipitation distribution and the prevalence of temporal mismatches between estimated and observed precipitation peaks. The problem is illustrated in the paper by Figure 3, which shows a higher RMSE value (i.e., "worse" performance) for the bias- and CDF-corrected SM2RAIN product (BC-CDF) than for any of the uncorrected SM2RAIN products.*

We agree with the reviewer that the RMSE statistic has some limitations in evaluating precipitation datasets. Indeed, we have used different statistics and in the revised manuscript we have performed the evaluation with multiple statistics also for Figure 6. Anyhow, RMSE is used in many papers evaluating precipitation datasets (and we don't believe they are all wrong), and it suffers from the same limitation of any single score; an assessment by using multiple scores is needed. As mentioned above, we believe that the real validation should be performed using the rainfall products in the hydrological or agricultural applications. These aspects have been underlined in the revised manuscript as shown in the reply of comment (3).

*(4) Only correlation and RMSE statistics are presented for the performance evaluation in Figure 6. Please remove the RMSE for the previously mentioned reason and add other metrics, such as variability ratio, bias, hit/miss ratio, frequency of wet days, peak magnitude, etc. for a more thorough performance evaluation.*

In the revised manuscript, we have added multiple statistics in Figure 6, similarly to Figure 3.

*(5) The TC evaluation only takes into account the monthly correlation – just one aspect of dataset performance (monthly temporal dynamics). Hence the TC evaluation alone cannot be used to conclude*

*whether a particular dataset is better or worse (as is done in the last paragraph of the abstract: "SM2RAIN-ASCAT dataset provides better performance better than GPM and GPCC in the data scarce regions of the world"). Other aspects should also be considered.*

TC analysis is performed at daily time scale, not monthly time scale. Therefore, we believe TC analysis provides information on the accuracy of the different rainfall products at daily time scale, it has been clarified in the revised manuscript at lines 507-509:

"On a global scale, the TC approach has been implemented by using the triplet SM2RAIN-ASCAT, GPM-ER and GPCC, by considering the common period 2015-2018 and at daily time scale."

*(6) "The recent "bottom up" approach that uses satellite soil moisture observations for estimating rainfall through the SM2RAIN algorithm is suited to build long-term and consistent rainfall data record as a single polar orbiting satellite sensor is used." If this is true, why does the dataset span such a short period (2007-2018)? All datasets listed in Table 1 (excluding IMERG) span a longer period. This statement should be revised.*

The statement has been revised as we intended to say that a long-term SM2RAIN-ASCAT dataset, starting from 2007, and ensured until mid-2040s, can be built based the proposed approach. Sorry for the misunderstanding that has been corrected in the revised manuscript (see lines 29-33):

"We exploit here the Advanced SCATterometer (ASCAT) on board three Metop satellites, launched in 2006, 2012 and 2018, as part of the EUMETSAT Polar System programme. The continuity of the scatterometer sensor is ensured until mid-2040s through the Metop Second Generation Programme. By applying SM2RAIN algorithm to ASCAT soil moisture observations, a long-term rainfall data record will be obtained, starting in 2007 until mid-2040s."

*(7) On a related note, the evaluation of https://www.hydrol-earth-systsci.net/21/6201/2017/ (co-authored by the first author of the present study) shows that SM2RAIN-ASCAT performs worst among all precipitation datasets in terms of trend, due to the combination of data from different ASCAT sensors. So are the different ASCAT sensors consistent with each other or not? Has this trend issue been resolved in this SM2RAIN-ASCAT release? If so, this should be shown. If not, this should be communicated to the reader.*

The trend issue has been solved as in the previous delivery of the SM2RAIN-ASCAT dataset (preliminary distribution) we did not consider appropriately the availability of two ASCAT sensors (Metop-A and -B) after 2013. The dual calibration performed in this study (see lines 448-450) has been carried out exactly to address this issue. It has been clarified in the revised manuscript at lines 450-452:

"The dual calibration has solved the issue in terms of long-term trend that has been found in previous application of SM2RAIN to ASCAT soil moisture data (Beck et al., 2017)."

*(8) In the interest of transparency the abstract should mention that the presented SM2RAIN dataset i) is limited to liquid precipitation (snowfall is not present in the dataset), ii) exhibits spurious drizzle, iii) underestimates extremes (as demonstrated by Figures 3 and 4 of the paper), and iv) potentially suffers*

*from intercalibration issues (see comment (7)). If any of these problems have been fixed in the current release of SM2RAIN-ASCAT, this should be shown in the paper.*

As suggested by the reviewer, we have clearly communicated the limitations of SM2RAIN-ASCAT dataset in the abstract of the revised manuscript. Limitations and strengths of the SM2RAIN-ASCAT dataset have been clearly illustrated (see replies to comment (1)). In the interest of transparency, we have made the SM2RAIN-ASCAT product freely available, and also the dataset at 1009 points that we have used for selecting the best configuration to develop the product. All the input and test datasets used in the paper are freely available and the analysis can be easily performed by the reader (note that also SM2RAIN code is made available on GitHub).

*(9) "The limitations of the bottom up approach are the possibility to estimate only terrestrial rainfall and its dependence on land characteristics (e.g., low accuracy for dense vegetation coverage and complex topography, Broccaet al., 2014)." The other limitations (spurious drizzle, underestimation of extremes, and intercalibration issues) should also be mentioned here.*

Limitations and strengths of the SM2RAIN-ASCAT dataset have been clearly illustrated in the revised manuscript (see replies to comment (1)).

*(10) To my understanding the regional evaluation is performed using daily accumulations, while the triple collocation (TC) analysis is performed using monthly accumulations – correct? To avoid confusion, please state the time scale of each specific evaluation/analysis in both the abstract and the captions of all figures.*

All the analyses have been performed at daily time scale and it has been clarified in the revised manuscript (lines 507-509):

"On a global scale, the TC approach has been implemented by using the triplet SM2RAIN-ASCAT, GPM-ER and GPCC, by considering the common period 2015-2018 and at daily time scale."

*(11) Version numbers should be assigned to the different SM2RAIN-ASCAT releases, to avoid confusion. I know there have been at least two releases. Which one is this?*

The first official version of the SM2RAIN-ASCAT dataset should be considered the one presented in this paper. Indeed, the dataset has been published on Zenodo and a DOI (digital object identifier) has been assigned to the dataset to avoid confusion.

*(12) Please add ERA5 to Figure 3 and make it easier to see the differences among the boxes, either by reducing the range of the y-axes or by exanding the size of the y-axes.*

In the revised manuscript, we have added ERA5 and we have also improved figure readability.

*(13) The intro/methods part of the abstract is a bit too long, while the results/discussion part is a bit too short (just three sentences).*

The abstract of the revised manuscript has been revised accordingly.

*(14) "the surface runoff rate, i.e., the water that does not infiltrate into the soil and flows at the surface to the watercourses, is much lower than the rainfall rate, mainly if equation (1) is applied at coarse spatial resolution (20 km), i.e., with satellite soil moisture data." This statement does not make sense to me. Runoff can be equal to rainfall if the soil is saturated, at all scales – from hillslope to catchment.*

With the runoff assumption, we are saying that surface runoff is expected to be negligible at larger spatial scales due to the possibility that locally generated surface runoff (e.g., over impervious surfaces) can re-infiltrate into more permeable areas in the same pixel. Of course, this hypothesis can be not valid in some areas, but we have indirectly validated this hypothesis as we have hardly seen the ASCAT soil moisture signal to be saturated for more than one day. Therefore, surface runoff due to saturated soil is expected to occur very rarely at 20 km scale. This aspect has been clarified in the revised manuscript showing the number of days the ASCAT soil moisture signal is saturated for more than one day (see lines 269-277):

"We have indirectly tested this hypothesis by counting the number of days the ASCAT soil moisture product is higher than 99.5 percentile for two (or more) consecutive days in the period 2007-2018. We have found that the number of consecutive days in which the soil is saturated is equal to 4 days (median value on a global scale) over 12 years, with 90% of land pixels with values lower than 12 days (i.e., 1 day per year). The occurrence of higher values is limited to very few areas in the tropical forests and over Himalaya (see Figure A2)."

[Figure]

**Figure A2.** Number of days in which ASCAT soil moisture observations are close to saturation (>99.5 percentile, top panel) for 2 (or more) consecutive days in the period 2007-2018. Please note that the upper value is set to 20 days as in most of land areas the occurrence is very low (90% of land pixel with values lower than 12 days over 12 years). In the bottom panel the soil moisture values at 99.5 percentile (in the period 2007-2018) are shown.

**Anonymous Referee #3**

*This study provides the descriptions and validation results of the 11-year (2007-2018) SM2RAIN-ASCAT global rainfall dataset. Overall, the study fits the scope of ESSD, the paper is well written and the presentation quality is very good. I think that the dataset has great potentials for different applications, as also stated by the authors, especially in specific regions of the world where it seems to outperform other products based on different approaches (e.g., GPM IMERG Early Run). However, there are several aspects that need to be addressed before the paper can be published.*

We thank the reviewer for her/his appreciation of our study and for the valuable suggestions that helped us to clarify and improve the manuscript. A detailed answer to each comment is reported in the sequel.

*Line 34: What do the author mean by "operationally available in NRT"? This is not a crucial aspect for the dataset presented in this paper.*

The reviewer is right; the sentence has been removed by the abstract of the revised manuscript. It is due to a parallel activity we are performing for producing a NRT SM2RAIN-ASCAT product.

*Line 36: It is important to note that it is not global as it does not provide rainfall over water bodies, and it is limited to the avalability and quality of soil mosture data. This should be clearly stated also in the conclusions.*

The reviewer is right; SM2RAIN-ASCAT product is not global and it has been clarified in the abstract and in the conclusions of the revised manuscript (see lines 34-36):

"The paper describes the recent improvements in data pre-processing, SM2RAIN algorithm formulation, and data post-processing for obtaining the SM2RAIN-ASCAT quasi-global (only over land) daily rainfall data record at 12.5 km sampling from 2007 to 2018."

And lines 552-555:

"Limitations of SM2RAIN-ASCAT data record consist in: 1) the underestimation of peak rainfall events, 2) the presence of spurious rainfall events due to high frequency soil moisture fluctuations, 3) the estimation of liquid rainfall only (snowfall cannot be estimated), and 4) the possibility to estimate rainfall over land only."

*Line 42: Please, specify "the IMERG Global Precipitation Measurement (GPM) mission product*

The text has been modified, accordingly (see lines 38-42):

"Moreover, an assessment on a global scale is provided by using the Triple Collocation technique allowing us also the comparison with the latest ECMWF reanalysis (ERA5), the Early Run version of the Integrated Multi-Satellite Retrievals for Global Precipitation Measurement (IMERG), and the gauge-based Global Precipitation Climatology Centre (GPCC) products."

*Line 75: Rainfall is not "measured" from space. Precipitation retrieval based on "topdown" approaches is very complex due to interaction of the radiation emitted by the Earth's surface with gases, and liquid*

*and solid hydrometeors within the clouds. For example, passive microwave retrieval techniques need to account for variability of all these elements (e.g., surf. emissivity and temperature, water vapour content, cloud water content, sizes, shapes, density, 3-D distribution of liquid, solid and mixed-phase hydrometeors).*

The term "measured" will be removed from the revised manuscript, even though it's matter of terminology. Every measurement is affected by errors. The text has been changed (see lines 76-78):

"The standard methods for estimating rainfall from space are based on instantaneous measurements obtained from microwave radiometers, radars, and infrared sensors (Kidd and Levizzani, 2011)."

*Line 76-78: Please, rephrase this sentence: "these methods are based on inversion techniques where the upwelling radiation (or backscattered signal for radars) is related to the surface precipitation rate".*

Accordingly, the sentence has been revised (see lines 78-81):

"These methods are based on inversion techniques where the upwelling radiation (or backscattered signal for radars) is related to the surface precipitation rate, i.e., a "top down" approach (Brocca et al., 2014)."

*Line 133 (and Line 176, and Line 327): Please, clarify what you mean by "1009 points". Are these 12.5km x1.2km grid boxes? What do you mean by "uniformly distributed? How have they been selected? How many "points" are selected in each region? How are the raingauge measurements treated to be associated to each "point"?*

The 1009 points are uniformly distributed over a regular grid with spacing of 1.5°. Each point is considered representative of a 0.25° x 0.25° box; the selection is carried out for reducing the computational time for running the different SM2RAIN configurations. The numbers of points for each region is based on the size of the region (328 points in Australia, 163 in India, 55 in Italy, and 463 in the United States). Ground observations and GPM-ER data are regridded by spatial averaging measurements contained over each 0.25° x 0.25° box. All these details have been reported in the revised manuscript (see lines 365-371):

"We have selected 1009 points uniformly distributed over a regular grid with spacing of 1.5°. Each point is considered representative of a 0.25° x 0.25° box. The selection is carried out for reducing the computational time in running the different SM2RAIN configurations. The numbers of points for each region is depending on the size of the region: 328 points in Australia, 163 in India, 55 in Italy, and 463 in the United States. Ground observations, GPM-ER and ERA5 data are regridded by spatial averaging measurements contained over each 0.25° x 0.25° box."

*Line 203-205: it is not clear how the 12 hour sampling of the ASCAT soil moisture product is used to obtain the daily (24 hour) SM2RAIN rainfall prouct.*

The 24-hour accumulated rainfall is obtained by summing the two 12-hour accumulated rainfall data obtained for each day, it has been specified in the revised manuscript at lines 227-229:

"The 24-hour accumulated rainfall is obtained by summing the two 12-hour accumulated rainfall data obtained for each day."

*Line 282-284: correction of the overall bias can be very effective for mitigating errors in all products. It should be pointed out by the authors if SM2RAIN-ASCAT dataset presented in this paper is the same product that would be obtained operationally in NRT (see also Line 34). If this is not the case, in my opinion, for a fair comparison, the IMERG GPM Final Run should be used instead of the Early Run in this study. Otherwise, the authors should explain clearly why the GPM Early Run is used in this study. Although I understand that IMERG Final Run can not be used for TC, I recommend to show the results of SM2RAIN-ASCAT compared to IMERG Final Run.*

The SM2RAIN-ASCAT dataset presented in the paper is the same product that would be obtained operationally in NRT. The climatological correction is performed with constant parameter values and, hence, it can be implemented in NRT. We note that a climatological correction is performed in several satellite rainfall datasets delivered in NRT (e.g., 3B42RT, IMERG ER, PERSIANN CCS, CMORPH CRT). It has been clarified in the revised manuscript at lines 319-322:

"Specifically, we refer here to a static correction procedure that once calibrated for a time period can be applied in the future periods, also for operational real time productions. We note that a climatological correction is performed in several satellite rainfall datasets delivered in near real-time (e.g., GPM Early Run)."

*Line 314: Optimal value for FAR is 0, not 1. Please, correct.*

The reviewer is right; we have corrected the error in the revised manuscript, thanks for spotting the mistake.

*Line 316: Please, motivate the choice of 0.5 mm/day (and not a lower value > 0 mm/day) as rainfall/no rainfall threshold.*

As mentioned in the manuscript, the threshold is selected in order to exclude spurious events that might be due to rainfall interpolation\regridding in the reference datasets.

*Line 379-380: How many points are used to compute these averages in each region? Are "problematic" areas for soil moisture retrieval (complex orography, highly vegetated, ecc.) included among the 1009 points used here?*

All points in each region are used, i.e., 328 points in Australia, 163 in India, 55 in Italy, and 463 in the United States. The "problematic" areas are included as 1009 points are randomly selected; no masking has been carried out in this analysis.

*Line 389: Why R and RMSE are considered "more important"? Please, justify this choice.*

We believe that R and RMSE are the two most important statistics for evaluating precipitation datasets after performing several assessment studies of different datasets. However, we acknowledge that the selection of the statistics could be arbitrary and in the revised manuscript we have added multiple statistics at Figure 6 (similarly to Figure 3) to provide a more comprehensive assessment of the products.

*Line 400-401: It is not clear what periods is used for the calibration in the two separate time frames. I assume that the calibration is not carried out for the whole periods.*

In the development of the global SM2RAIN-ASCAT dataset the calibration is performed for the whole periods. Indeed, we do not want to perform calibration and validation against ERA5. As mentioned above, the validation should be performed with independent datasets, and even better by using the product for applications.

*Line 404-408: it is not clear what the authors mean by distinguishing "in space" and "in time".*

In space, we mean a fixed spatial mask over which we are aware of the lower performance of the ASCAT soil moisture product, and consequently of SM2RAIN-ASCAT rainfall product. In time, we have considered a temporally variable mask that flags observations with soil temperature, obtained from ERA5, lower than 3°C. It has been specified better in the revised manuscript (see lines 453-459):

"In space (i.e., a fixed spatial mask), we have used the committed area mask developed for ASCAT soil moisture product (PVR 2017), a frozen probability mask and a topographic complexity mask. In time (i.e., a temporally variable mask), we have considered the soil temperature data from ERA5 and flagged the observations with soil temperature values between 0°C and 3°C as temporary frozen soil and below 3°C as frozen soil. As many applications require continuous data, we have preferred to flag the data instead of removing them with an expected loss of accuracy."

*Line 411-413: ERA-5 is used for calibration. It is not fair to use this dataset to create this map, and show R and RMSE.*

The reviewer is right; Figure 5 shows the consistency of ERA5 and SM2RAIN-ASCAT and not the "accuracy" or the "performance" of the product, these terms has been removed from this section of the revised manuscript. Of course, we expect better performance in the areas in which the consistency is higher, but the preliminary assessment of SM2RAIN-ASCAT is performed in section 4.3. See lines 462-466:

"Figure 5 shows R and RMSE values between SM2RAIN-ASCAT and ERA5 in a single map. Therefore, Figure 5 illustrates the consistency between SM2RAIN-ASCAT and ERA5, and it is not intended to assess the performance of the data record (even though we expect better accuracy in areas where the agreement is higher)."

*Line 468: Please, specify what is the committed area for ASCAT products (not ASCAT).*

The reviewer is right; the committed area refers to the ASCAT soil moisture product and not to ASCAT.

The committed area has been built from ASCAT soil moisture product developers to indicate the areas in which the quality of soil moisture retrieval is expected to be good; it has been specified in the revised manuscript (see lines 453-455):

"we have used the committed area mask developed for the ASCAT soil moisture product (i.e., the area in which the ASCAT soil moisture retrievals are expected to be good, PVR 2017)"

*Minor corrections:*

*Line 46: correct: "provides better performance better"*

The text has been modified, accordingly.

*Line 100: correct "has the advantage of requiring"*

The text has been modified, accordingly.

*Line 138-139: please specify which datasets have been used for the TC, what for the regional assessment, and what for global assessement.*

The datasets used for the three analyses has been specified in the revised manuscript (see lines 146-153):

"As reference datasets we have used high-quality local raingauge networks from 2013 to 2017 in the United States, Italy, India and Australia for the assessment at 1009 points and for the regional assessment. Three additional global datasets have been considered: the latest reanalysis of the European Centre for Medium-Range Weather Forecasts (ECMWF), ERA5, the gauge-based Global Precipitation Climatology Centre (GPCC), and the GPM IMERG product (Early Run version). ERA5 has been used for the generation of the quasi-global SM2RAIN-ASCAT data record; GPCC and GPM IMERG have been considered for the TC analysis."

*Line 190: Please, correct: "spatially averaging"*

The text has been modified, accordingly.

*Line 392-393: please correct this sentence. Something is missing, or maybe remove "," after "filtering".*

The sentence has been revised, see lines 440-443:

"As reference dataset for the calibration of the parameter values of the pre-processing (filtering), of SM2RAIN, and of the post-processing, the ERA5 rainfall has been used mainly because of its higher spatial resolution compared to GPCC (36 km versus 100 km)."

*Line 402-403: Please, correct this sentence.*

The sentence has been corrected, see lines 452-453:

[revised manuscript text omitted]